# TOWARDS HUMAN-LEVEL REASONING BENCHMARKS FOR MULTIMODAL LANGUAGE MODELS

## ABSTRACT

The goal of achieving Artificial General Intelligence (AGI) is to imitate humans and surpass them. Models such as OpenAI's o1, o3, and DeepSeek's R1 have demonstrated that large language models with human-like reasoning capabilities exhibit exceptional performance and are being gradually integrated into multimodal large language models. However, whether these models possess capabilities comparable to humans in handling reasoning tasks remains unclear at present. In this paper, we propose Human-Aligned Bench, a benchmark for fine-grained alignment of multimodal reasoning with human performance. Specifically, we collected 9,794 multimodal questions that solely rely on contextual reasoning, including bilingual (Chinese and English) multimodal questions and pure text-based questions, encompassing four question types: visual reasoning, definition judgment, analogical reasoning, and logical judgment. More importantly, each question is accompanied by human success rates and options that humans are prone to choosing incorrectly. Extensive experiments on the Human-Aligned Bench reveal notable differences between the performance of current MLLMs in multimodal reasoning and human performance. The findings on our benchmark provide insights into the development of the next-generation models.

## 1 INTRODUCTION

Contextual reasoning is fundamental to human intelligence (Sternberg, 1982; Lohman & Lakin, 2011), and also a goal for achieving Artificial General Intelligence (AGI) (Morris et al., 2023). Recent advancements in Large Language Models (LLMs) have demonstrated substantial improvements in reasoning capabilities across complex domains such as mathematics (Peng et al., 2025; Yang et al., 2024; Xu et al., 2024; Meng et al., 2025), logical reasoning (Wan et al., 2024; Xu et al., 2023; Feng et al., 2023; Liu et al., 2023a) and coding (Ahmad et al., 2025; Jiang et al., 2024b; Li et al., 2025; Huang et al., 2023). Techniques like test-time compute scaling (e.g., OpenAI o1 (Jaech et al., 2024) and Deepseek-R1 (Guo et al., 2025a)) have significantly enhanced the reasoning performance of LLMs (Guo et al., 2025a; Goertzel & Pennachin, 2007; Wang et al., 2024c), while gradually becoming more human-like in their reasoning forms. With the rapid advancement in research on language reasoning of LLMs (Yang et al., 2025; Peng et al., 2025; Meng et al., 2025; Chen et al., 2025; Liu et al., 2025b; Wang et al., 2025; Liu et al., 2025a; Shen et al., 2025; Wang & Peng, 2025), investigations into reasoning multimodal large language models (MLLMs) have emerged as a dominant research direction.

However, current tasks predominantly fall into knowledge-intensive categories that require domain-specific expertise beyond substantial reasoning capabilities in fields such as science or mathematics, making it challenging to isolate and evaluate the contextual reasoning capacities of MMLMs. Although some researchers have proposed tasks relying solely on general reasoning skills to solve problems (Song et al., 2025; Cai et al., 2025; Xu et al., 2025), these studies still exhibit several limitations: *1)* Overemphasis on visual reasoning while neglecting text-based reasoning. *2)* Insufficient fine-grained annotations, particularly precise human performance data and solution methodologies for specific problem types. *3)* Absence of systematic analysis distinguishing whether MLLMs are merely memorizing procedural steps or engaging in genuine reasoning processes. These deficiencies hinder comprehensive evaluation of reasoning-oriented MLLMs' capabilities in multimodal scenarios and limit exploration regarding whether MMLMs' reasoning patterns align with human cognitive processes.

Table 1: **Comparison between our Human-Aligned Bench and existing multimodal contextual reasoning benchmarks (MM-IQ (Cai et al., 2025), VisuLogic (Xu et al., 2025), VISUALPUZ-ZLES (Song et al., 2025)).** Human-Aligned Bench includes more comprehensive data and question coverage. The systematic knowledge structuring and dynamic test-time augmentation also provide more reliable and fair evaluation of MLLMs. T: Text; I: Image; HAR: Human Accuracy Rates; HEO: Human Error-prone Options. HS: Human Solutions

| Benchmarks | Data Coverage | | Modality | HAR | HEO | HS |
| | #Instances | #Images | | | | |
| --- | --- | --- | --- | --- | --- | --- |
| MM-IQ | 2,710 | 2,710 | I+T | ✗ | ✗ | ✗ |
| VisuLogic | 1,000 | 1,000 | I+T | ✗ | ✗ | ✗ |
| VISUALPUZZLES | 1,168 | 1,168 | I+T | ✗ | ✗ | ✗ |
| Human-Aligned Bench | **9,794** | **2,759** | **T, I+T** | ✔ | ✔ | ✔ |

To address these challenges, we introduce Human-Aligned Bench, a benchmark designed to evaluate the pure reasoning capabilities of MLLMs. The data for Human-Aligned Bench is primarily sourced from the logical reasoning and judgment sections of China's civil service examinations, which mainly assess test-takers' abilities in contextual understanding and logical thinking without requiring any additional prior knowledge. Thus, these types of questions serve as an ideal testing platform for reasoning multimodal models. Specifically, as shown in Table 1 and Fig. 1, our Human-Aligned Bench comprises 9,794 reasoning problems spanning four distinct reasoning categories: Visual Reasoning, Definition Judgment, Analogical Reasoning, and Logical Judgment. This benchmark supports bilingual (Chinese-English) visual-textual and text-only question-answering formats. For each question, we also compiled human accuracy rates and pinpointed the distractor choices that most frequently mislead respondents. Moreover, for every question category, we offer concise synopses of human solution strategies—serving as a foundation for multidimensional analysis and cross-validation of MLLMs' inferential processes, thereby illuminating their genuine reasoning prowess. Compared with existing reasoning benchmarks, our Human-Aligned Bench is expected to provide a more systematic evaluation in exploring whether the contextual reasoning abilities of MLLMs align with human reasoning.

We conducted comprehensive evaluations and systematic analyses using the Human-Aligned Bench to assess the reasoning capabilities of current MLLMs. Extensive experimental results show that large models trained with reasoning-related data (e.g., Gemini-2.5-pro-exp-03-25) generally outperform conventional smaller models in visual reasoning tasks, yet still exhibit a significant performance gap relative to human benchmarks. In textual reasoning tasks, all MLLMs demonstrated superior performance compared to their visual reasoning capabilities, with similar performance discrepancies observed in bilingual tasks, highlighting the persistent challenges in achieving robust visual reasoning within current multimodal architectures. Regarding human capability alignment, MLLMs failed to exhibit accuracy improvements corresponding to reduced task difficulty levels in image-based reasoning, though their error-prone patterns at high difficulty levels showed partial alignment with human cognitive tendencies. Furthermore, through joint analysis of the model's reasoning processes and human solutions, we found that certain MLLMs' inferential faculties remain dependent on predefined prompts and some MLLM rise fake reasoning. This critical dependency reveals fundamental disparities between the reasoning patterns of MLLMs and human cognitive processes. Our contributions are summarized as follows:

- **Multimodal Contextual Reasoning Benchmark.** We introduce Human-Aligned Bench, a multimodal benchmark that relies solely on contextual reasoning, designed to comprehensively evaluate the reasoning capabilities of MLLMs.

- **Analysis of alignment with human capabilities.** We conduct a fine-grained analysis of the anthropomorphic characteristics in MLLMs by utilizing human average accuracy rates and error-prone options.

- **Analysis of Fake Reasoning Abilities.** We conduct joint analysis using human's prior reasoning processes and the built-in reasoning of models to assess how MLLMs apply and learn reasoning processes.

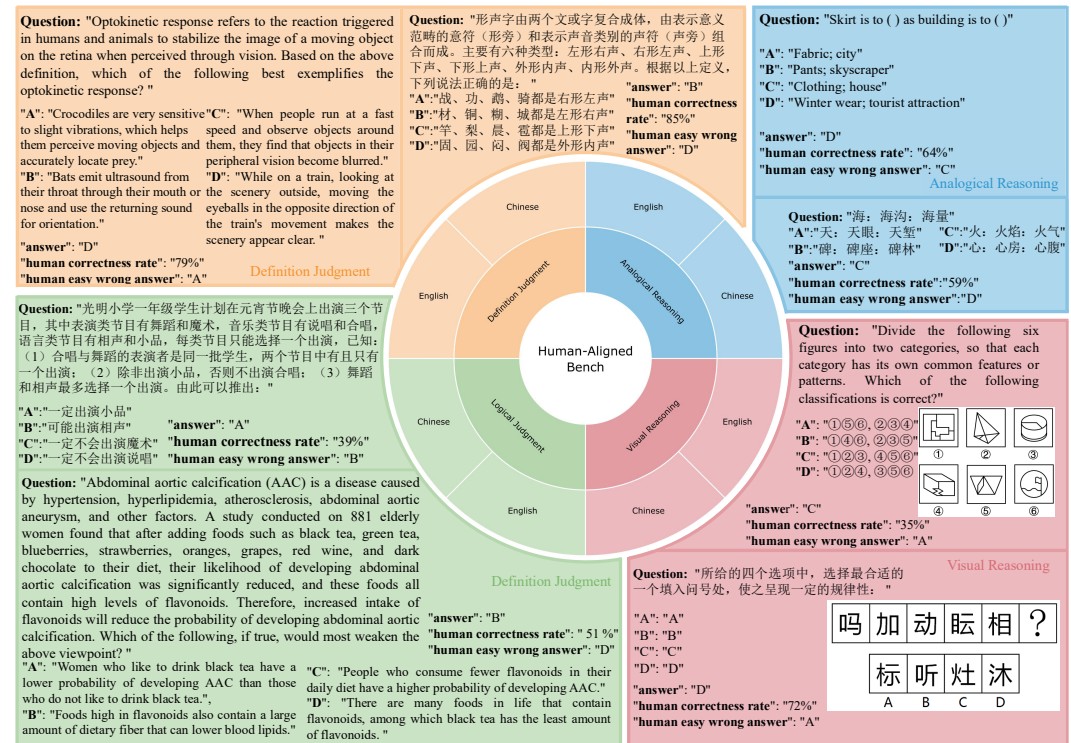

Figure 1: **Overview of Human-Aligned Bench.** Human-Aligned Bench contains 4 categories of questions, each of which has both Chinese and English versions. Each question contains human scoring rates and error-prone options. These questions require models' abilities in visual logic and pure text reasoning.

## 2 RELATED WORK

**Multi-modal Large Language Models.** The recent epoch has been distinguished by considerable strides in the development of MLLMs. This progression originated with foundational endeavors—such as BLIP (Li et al., 2022; 2023) and Flamingo (Alayrac et al., 2022)—which innovatively introduced lightweight adapters to bridge vision transformers (Dosovitskiy et al., 2020) and Large Language Models. Following these, instruction-tuned variants like LLaVA (Liu et al., 2024a) and MiniGPT-4 (Zhu et al., 2024) emerged, significantly augmenting multimodal perception. Concurrently, proprietary systems including GPT-4o (OpenAI, 2024), Gemini-Pro (Team et al., 2023), and Claude (Anthropic, 2024) have demonstrated exceptional performance on sophisticated multimodal benchmarks. In parallel, open-weight architectures—spanning the Qwen-VL series (Bai et al., 2023; Wang et al., 2024a; Bai et al., 2025) and InternVL variants (Chen et al., 2024b;c;a; Wang et al., 2024b; Zhu et al., 2025), to LLavaOV, Pangea, Cambrian, and various llama-based approaches (Li et al., 2024; Yue et al., 2025; Liu et al., 2024b; Tong et al., 2024; Dubey et al., 2024)—vie for prominence through refined architectural optimizations, strategic dataset expansions, and the implementation of novel training paradigms. Furthermore, domain-specific MLLMs have been conceptualized for specialized applications, leveraging techniques such as multimodal pretraining, vision instruction tuning, and reinforcement learning. Notable examples include Math-LLaVA (Shi et al., 2024) and MultiMath (Peng et al., 2024), designed for advanced mathematical reasoning, alongside Med-Flamingo (Moor et al., 2023), LLaVA-Med (Liu et al., 2024a), and Med-MoE (Jiang et al., 2024a) within the biomedical sphere. Recently, the burgeoning and rapidly advancing domain of multimodal reasoning has become the focus of intensive exploration. Pertinent research in this area has notably employed anthropomorphic chain-of-thought prompting (Wei et al., 2022), iterative guidance methodologies, and a suite of reinforcement learning-based techniques, including DPO (Zhong et al., 2024), STaR (Zelikman et al., 2024b), Quiet-STaR (Zelikman et al., 2024a), and DeepSeek-E1 (Guo et al., 2025a). Augmenting these approaches are specifically architected, reinforcement learning-driven models. This cohort includes R1-Onevision (Yang et al., 2025), LMM-R1 (Peng et al., 2025), MM-EUREKA (Meng et al., 2025), R1-V (Chen et al., 2025), Visual-rft (Liu

Figure 2: **Data curation and statistics of our Human-Aligned Bench.** The data curation pipeline consists of four stages: data collection, screening, parsing, and processing.

et al., 2025b), Visualprm (Wang et al., 2025), OThink-MR1 (Liu et al., 2025a), VLM-R1 (Shen et al., 2025), Open-r1-Video (Wang & Peng, 2025), QvQ (Team, 2024), Claude-3.7-Sonnet-thinking (Anthropic, 2024), o1 (Jaech et al., 2024), and Gemini-2.0-flash-thinking (Team et al., 2023). Notwithstanding its current nascent stage, these pioneering contributions have illuminated new investigative pathways towards the realization of AGI.

**Multimodal Reasoning Benchmarks**. With the rapid MLLMs, multimodal benchmark has evolved from early visual perception tasks to specialized domains, including OCR (Liu et al., 2024d), Chart Question Answering (Masry et al., 2022), Document Visual Question Answering (Mathew et al., 2021), Agent Benchmark (Liu et al., 2023b), Tool Vision Benchmark (Ye et al., 2024), and first-person perspective perception tasks (Mangalam et al., 2023; Cheng et al., 2024). While many benchmarks aim to evaluate MLLMs' general world knowledge and reasoning abilities (Yue et al., 2024a; Marino et al., 2019; Liu et al., 2024c; Yue et al., 2024b; Authors, 2025), and others like MathVerse (Zhang et al., 2024), MMBench (Liu et al., 2024c), GSM-8K (Cobbe et al., 2021), and EXAMS-V (Das et al., 2024) focus on domain-specific or exam-style challenges (Saikh et al., 2022; Lu et al., 2023; Meng et al., 2024), most benchmarks evaluate both model knowledge and reasoning faculties in tandem, without effectively isolating the latter for discrete assessment. EMMA (Hao et al.), its 14 K multimodal reasoning questions that require cross-modal deduction and the new chain-of-thought annotation protocol. RBench-V (Guo et al., 2025b), the first benchmark that allows free-form visual outputs (segmentation masks, sketches, plots) and the associated V-Score metric. Neither EMMA nor RBench-V supply human solution strategies per-question human accuracy rates or human error-prone options, which are necessary for fine-grained alignment analysis.

Recently, an increasing number of multimodal benchmark evaluations has turned its attention to contextual multimodal reasoning capabilities (Cai et al., 2025; Xu et al., 2025; Song et al., 2025). Nonetheless, these studies exhibit several critical limitations: **1)** A disproportionate emphasis is placed on visual reasoning, often at the expense of thorough exploration into text-based reasoning. **2)** There remains a paucity of fine-grained annotations, most notably the absence of precise human performance metrics tied to specific problem categories and solution strategies. **3)** There is a lack of systematic inquiry into whether MLLMs are genuinely performing reasoning or merely reproducing memorized procedural patterns. These deficiencies represent fundamental benchmarks for evaluating the authenticity of AGI.

## 3 HUMAN-ALIGNED BENCH

As shown in Fig. 2, The data curation of Human-Aligned Bench involves four stages:

**Data Collection.** We collected questions from the public examination papers of Chinese Civil Service Examination, which is held annually at the national and provincial levels, thus increasing the diversity of the data. The data collection involves an extensive search of online open-source exam repositories. Since millions sit for the civil service examination each year and most opting to prepare via online platforms that report human accuracy rates, human error-prone options, and each question has corresponding knowledge points, we have consolidated this information into the metadata repository of the Human-Aligned Bench.

**Data Screening.** Following the initial data collection, human annotators transcribed mathematical formulas, chemical expressions, and special symbols—originally stored in image format—into textual representations. Subsequently, we employed the MD5 hash algorithm to eliminate duplicate questions. Next, a team of data annotators categorized the question content into four distinct types—Visual Reasoning, Definition Judgment, Analogical Reasoning, and Logical Judg-

ment—guided by knowledge point identification. These classifications were then cross-validated to ensure consistency and accuracy. Finally, given that certain questions encompassed culturally specific elements—such as Chinese idioms or classical poetry—the annotators divided the dataset into two overarching categories: Chinese-specific and non-Chinese-specific questions. Specifically, two civil servants with master's degree who has already been employed were invited to identify specific questions related to China, based on whether the questions and options contained specific Chinese expressions such as Chinese idioms or poems. Considering that some images contain Chinese text, we employed image translation tools followed by manual verification, and accordingly categorized these questions as Chinese-specific. Since only a limited number of questions featured Chinese text within images, we proportionally selected additional questions—based on human accuracy rates—from those without Chinese image content to balance the dataset, and likewise designated them as Chinese-specific. Specifically, a master of arts in English and two civil servants with master's degree who has already been employed were invited to collaborate on the meticulous review on translation. First, the two civil servants cross-checked the test points of each question to verify whether the logical relationships, reasoning steps, and "traps" in the questions were consistent with the original Chinese text. Second, the master of arts in English and the two civil servants conducted a cross-review of the domain-specific terminology in the questions to confirm whether internationally recognized or industry-standard English expressions were adopted. They also examined whether the translated text was idiomatic and natural, eliminated all instances of "Chinglish," and ensured that the sentences were free of ambiguity-with particular attention to the clarity of pronoun references. Finally, all questions containing translation issues were selected and manually translated by the meticulous review team.

**Data Parsing.** After the screening stage, each question is transformed into a standardized format comprising the question stem, options, answers, human correct rate, human error-prone options, question type and images (if any), and store all the information systematically.

**Data Processing.** Upon completing data parsing, we conducted a series of post-processing steps to ensure structural consistency across all questions. Leveraging the GPT-4o API, we translated all non-Chinese-specific questions into English, followed by meticulous review by domain experts to confirm technical accuracy. Furthermore, in collaboration with individuals experienced in civil service examinations, we developed problem-solving frameworks categorized by question type, linking each question to its corresponding framework to enhance the contextual foundation for reasoning.

**Data Statistics.** As presented in Table 2, the Human-Aligned Bench comprises 9,794 instances, with a near-uniform distribution across the four reasoning categories, ensuring that no single type disproportionately influences the benchmark. Likewise, difficulty levels—determined by human accuracy rates—are evenly balanced to accommodate a broad spectrum of cognitive demands. The dataset is linguistically diverse, with approximately half of the questions presented in Chinese and the

Table 2: Statistics of Human-Aligned Bench. HCR: Human Correct Rate

| Category | Statistics |
|---|---|
| Total Questions | 9794 |
| - Visual Reasoning | 2759 |
| - Definition Judgment | 2361 |
| - Analogical Reasoning | 2407 |
| - Logical Judgment | 2267 |
| HCR (0-60%, 60-80%, 80-100%) | 30%/35%/35% |
| Question Type (English/Chinese) | 51% / 49% |

other half in English, allowing for the assessment of models' reasoning capabilities across languages. Recognizing that robust MLLMs must also contend with purely textual inputs, the Human-Aligned Bench integrates both unimodal and multimodal instances.

## 4 EXPERIMENTS AND RESULTS

### 4.1 EXPERIMENTAL SETUP

We selected a diverse set of proprietary and open MLLMs to ensure broad coverage in terms of model architecture, training scale, and intended application domains. This diversity allows us to capture a wide spectrum of current approaches and capabilities in the field.

**Open Models.** We further evaluate widely used open MLLMs to gauge how open models compare against proprietary models, Intern-VL3-28B, Intern-VL3-78B (Zhu et al., 2025), Qwen2-VL-72B-

Instruct, Qwen2.5-VL-72B-Instruct, and QvQ-72B-Preview (Yang et al., 2024; Bai et al., 2025; Team, 2024).

**Proprietary Models.** We evaluate several leading proprietary models that represent the current state of the art: GPT-4o (OpenAI, 2024), o4-mini (OpenAI, 2025), Gemini-2.0-Flash, Gemini-2.0-Flash-Thinking, Gemini-2.5-Pro (Team et al., 2023; Gemini, 2025), Claude-3.7-Sonnet-thinking (Anthropic, 2024), and QvQ-Max (Qwen, 2025).

For open-source models and proprietary models, we deploy their weights locally, setting the temperature to 0.6 while keeping all other parameters at their default values. When designing the questioning template, we inform the MLLMs that they will subsequently encounter tasks in four categories: Visual Reasoning, Definition Judgment, Analogical Reasoning, and Logical Judgment.

## 4.2 OVERALL RESULT

In this subsection, we compare the performance of 11 opensource and proprietary models. Given the multiple dimensions of our benchmark, we analyze the MLLMs results from the reasoning type, language type, and modality type of the questions. The results are presented in Table 3 and Fig 3.

**Compare on Reasoning Type.** Visual reasoning evaluates abstract cognitive abilities by requiring individuals to discern visual patterns and infer missing components in accordance with underlying structural principles. Definition judgment, by contrast, assesses the precision of conceptual understanding, necessitating strict conformity to predefined criteria to determine whether candidate options fulfill the definitions articulated in the question stem. Analogical reasoning gauges the ability to recognize and align logical relationships between entities, demanding the identification of structural correspondences analogous to those presented in the prompts. Logical judgment appraises critical thinking and inferential reasoning skills, encompassing the evaluation of argument validity through analysis of premises and conclusions, as well as the derivation of conclusions via deductive or inductive logic. To comprehensively assess the multifaceted reasoning capacities of various MLLMs, we employ these four categories. As shown in Table 3, Gemini-2.5-pro-exp-03-25 surpassed other models in analogical reasoning, definition judgment, logical judgment and visual reasoning tasks. QvQ-Max attained the second high performance among all MLLMs in both analogical reasoning and definition judgment tasks. Meanwhile, the o4-mini also demonstrates excellent performance in logical reasoning, while Claude-3-7-sonnet-20250219-thinking surpasses all other models except Gemini-2.5-pro-exp-03-25 in visual reasoning. Notably, performance in analogical reasoning lagged markedly behind that of definition and logical judgment, highlighting critical limitations in current MLLMs: (1) deficient semantic comprehension of task instances; (2) limited capacity for modeling complex logical relations; and (3) underdeveloped ability to generalize or map relational structures across analogous contexts.

**Compare on Language Type.** The Fig. 3 (a) presents a comparative summary of the performance of various MLLMs on reasoning tasks formulated in both English and Chinese. Broadly speaking, all evaluated models exhibit superior accuracy on English-language questions—a disparity likely stemming from the predominance of English data in their training corpora and indicative of underlying challenges in cross-lingual multimodal reasoning. In the English subset, gemini-2.5-pro-exp-03-25 secures the highest score, surpassing the second-best model by a margin of 2.79%. For the Chinese tasks, qvq-max narrowly outperforms o4-mini by 0.37%, thereby establishing itself as the leading model in Chinese-language reasoning.

**Compare on Modality Type.** The Fig. 3 (b) reveals that the evaluated MLLMs demonstrate markedly lower accuracy on image-based inquiries in comparison to their efficacy in pure text reasoning tasks. This observation is consistent with existing benchmark results. Moreover, as shown in Table 6, although MLLMs demonstrate comparatively stronger proficiency in text-based reasoning within multimodal contexts than in image reasoning, they continue to underperform relative to specialized LLMs in dedicated textual reasoning settings. These findings indicate that MLLMs still have significant room for improvement not only in image reasoning but also in pure text reasoning.

## 4.3 COMPARE WITH HUMAN ABILITY

Considering that our Human-Aligned Bench includes human accuracy rates and error-prone options commonly made by humans, it can evaluate whether MLLMs exhibit reasoning performance aligned

Table 3: **Performance (%) on the four reasoning tasks (Analogical Reasoning, Definition Judgment, Logical Judgment, Visual Reasoning) of the Human-Aligned Bench across five human difficulty levels.** The table shows the evaluation scores of MLLMs, reflecting their average accuracy across different reasoning capabilities and difficulty levels. Top performers in each category are highlighted in **bolded**.

| Human | 0-20% | 20-40% | 40-60% | 60-80% | 80-100% | Overall |
|---|---|---|---|---|---|---|
| **Analogical Reasoning** | | | | | | |
| Claude-3-7-sonnet-20250219-thinking | 24.07 | 39.61 | 52.62 | 58.31 | 71.30 | 58.45 |
| Gemini-2.0-flash | 16.67 | 36.47 | 44.76 | 53.27 | 68.05 | 53.51 |
| Gemini-2.0-flash-thinking-exp | 27.78 | 33.73 | 45.69 | 55.42 | 68.57 | 54.55 |
| Gemini-2.5-pro-exp-03-25 | **42.59** | **54.90** | **65.92** | **74.43** | **88.70** | **74.32** |
| GPT-4o | 29.63 | 31.76 | 41.20 | 51.13 | 66.62 | 51.35 |
| o4-mini | 27.78 | 37.25 | 56.74 | 66.37 | 81.69 | 65.18 |
| QvQ-Max | 37.04 | 44.49 | 58.83 | 68.73 | 82.05 | 67.53 |
| InternVL3-38B | 22.22 | 40.39 | 55.43 | 65.49 | 80.65 | 64.48 |
| InternVL3-78B | 24.07 | 37.65 | 51.12 | 63.48 | 80.00 | 62.40 |
| QvQ-72B-Preview | 37.04 | 41.96 | 50.19 | 60.96 | 75.97 | 60.82 |
| Qwen2-VL-72B-Instruct | 27.78 | 33.73 | 48.13 | 55.79 | 74.94 | 57.25 |
| Qwen2.5-VL-72B-Instruct | 22.22 | 31.76 | 44.19 | 53.90 | 72.47 | 54.63 |
| **Definition Judgment** | | | | | | |
| Claude-3-7-sonnet-20250219-thinking | 15.38 | 55.04 | 66.47 | 80.82 | 93.93 | 83.19 |
| Gemini-2.0-flash | 7.69 | 50.39 | 67.64 | 80.16 | 93.39 | 82.59 |
| Gemini-2.0-flash-thinking-exp | 15.38 | 57.36 | 65.60 | 80.69 | 93.48 | 82.93 |
| Gemini-2.5-pro-exp-03-25 | **38.46** | **66.67** | **84.26** | **89.02** | **96.34** | **90.30** |
| GPT-4o | 38.46 | 47.29 | 64.43 | 81.35 | 94.91 | 83.23 |
| o4-mini | 30.77 | 58.91 | 73.76 | 88.10 | 96.25 | 87.97 |
| QvQ-Max | 38.46 | 60.47 | 77.26 | 87.95 | 96.34 | 88.60 |
| InternVL3-38B | 23.08 | 52.71 | 62.10 | 82.67 | 94.11 | 83.14 |
| InternVL3-78B | 15.38 | 60.47 | 69.39 | 85.19 | 96.34 | 86.45 |
| QvQ-72B-Preview | 23.08 | 53.49 | 70.85 | 81.22 | 93.12 | 83.52 |
| Qwen2-VL-72B-Instruct | 15.38 | 47.29 | 63.56 | 78.70 | 93.93 | 81.66 |
| Qwen2.5-VL-72B-Instruct | 15.38 | 48.84 | 67.93 | 83.60 | 95.36 | 84.63 |
| **Logical Judgment** | | | | | | |
| Claude-3-7-sonnet-20250219-thinking | 17.65 | 39.62 | 59.08 | 69.97 | 88.05 | 72.78 |
| Gemini-2.0-flash | 11.76 | 38.99 | 49.87 | 71.48 | 90.71 | 72.70 |
| Gemini-2.0-flash-thinking-exp | 23.53 | 46.54 | 56.52 | 77.39 | 90.49 | 76.44 |
| Gemini-2.5-pro-exp-03-25 | 23.53 | **61.64** | **75.96** | **86.56** | **94.80** | **85.80** |
| GPT-4o | 29.41 | 32.70 | 39.13 | 66.83 | 91.59 | 69.25 |
| o4-mini | **41.18** | 50.31 | 74.94 | 83.42 | 92.70 | 83.02 |
| QvQ-Max | 35.29 | 47.80 | 65.98 | 81.01 | 94.03 | 80.94 |
| InternVL3-38B | 29.41 | 33.33 | 43.22 | 69.97 | 90.27 | 70.58 |
| InternVL3-78B | 23.53 | 29.56 | 49.36 | 76.13 | 93.92 | 74.94 |
| QvQ-72B-Preview | 29.41 | 43.40 | 57.29 | 74.25 | 90.82 | 75.43 |
| Qwen2-VL-72B-Instruct | 11.76 | 25.79 | 42.20 | 67.09 | 88.94 | 68.20 |
| Qwen2.5-VL-72B-Instruct | 17.65 | 25.79 | 43.99 | 73.12 | 92.15 | 71.95 |
| **Visual Reasoning** | | | | | | |
| Claude-3-7-sonnet-20250219-thinking | 40.91 | **27.31** | 24.66 | 28.31 | 34.03 | 28.74 |
| Gemini-2.0-flash | 18.18 | 31.33 | 22.74 | 26.65 | 27.31 | 26.13 |
| Gemini-2.0-flash-thinking-exp | 22.73 | 22.89 | 25.48 | 24.26 | 25.67 | 24.79 |
| Gemini-2.5-pro-exp-03-25 | **45.45** | 24.10 | 27.67 | **29.50** | **35.97** | **30.23** |
| GPT-4o | 27.27 | 26.91 | 26.71 | 28.40 | 29.40 | 28.05 |
| o4-mini | 13.64 | 22.89 | 25.48 | 27.48 | 31.49 | 27.40 |
| QvQ-Max | 13.64 | 23.69 | 23.56 | 27.02 | 29.40 | 26.28 |
| InternVL3-38B | 18.18 | 24.50 | **28.77** | 27.11 | 29.40 | 27.80 |
| InternVL3-78B | 27.27 | 22.49 | 23.15 | 24.54 | 28.96 | 25.08 |
| QvQ-72B-Preview | 18.18 | 26.51 | 27.12 | 25.46 | 31.04 | 27.29 |
| Qwen2-VL-72B-Instruct | 13.64 | 21.29 | 26.71 | 27.21 | 35.37 | 28.42 |
| Qwen2.5-VL-72B-Instruct | 18.18 | 24.90 | 27.81 | 28.40 | 31.34 | 28.56 |

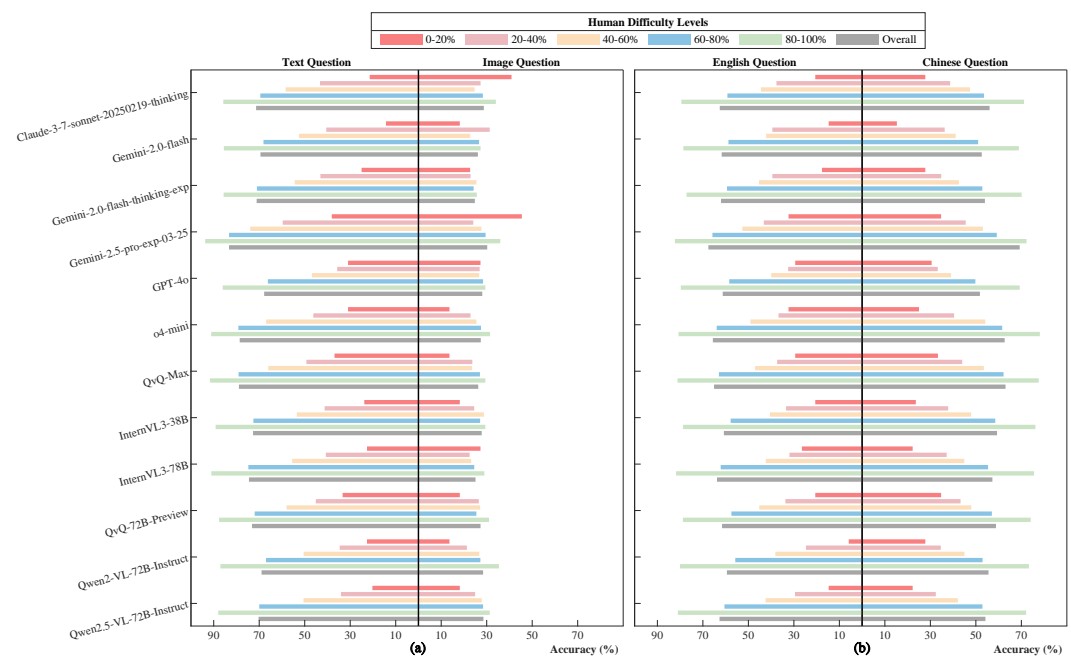

Figure 3: **Performance (%) on the Human-Aligned Bench across different modalities (text, image) and bilingual (Chinese and English ) in five human difficulty levels.** (a) delineates the evaluation metrics for MLLMs, showcasing their mean accuracy across different modalities and tiers of difficulty. (b) presents the evaluation metrics for MLLMs, illustrating their mean accuracy across different languages and levels of complexity.

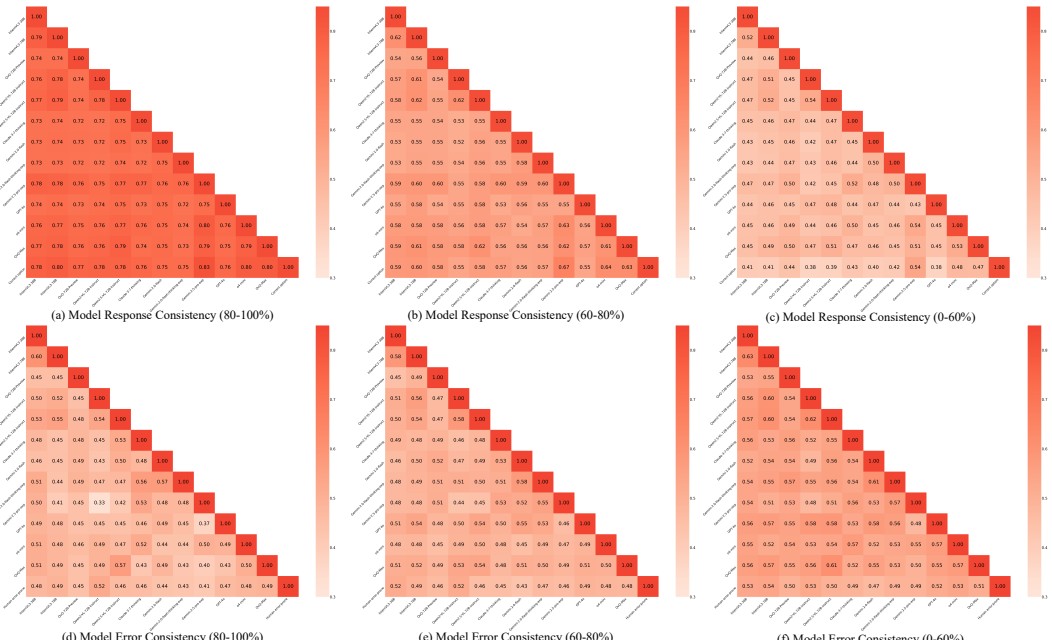

Figure 4: **Consistency on Response and Error.** (a), (b), and (c) represent the Consistency on Response between MLLMs and humans. (d), (e), and (f) denote the Consistency on Error between MLLMs and humans. In the heatmap, each position (x, y) represents the proportion of consistency between the results of prediction x and prediction y across all responses/all errors.

with humans. Therefore, in this subsection, we analyze the comparative results between MLLMs and humans from three perspectives: accuracy trends, consistency of model responses, and error consistency.

Table 4: **Performance (%) on the Human-Aligned Bench across different reasoning patterns and five human difficulty levels.** The table shows the evaluation scores of MLLMs, reflecting their average accuracy across different reasoning modes and difficulty levels. WHS: With human Solution; WSS: With self Solution.

| Human | 0-20% | 20-40% | 40-60% | 60-80% | 80-100% | Overall |
|---|---|---|---|---|---|---|
| InternVL3-78B | 23.58 | 34.97 | 43.69 | 58.85 | 79.04 | 60.59 |
| InternVL3-78B-WHS | 21.70 | 33.21 | 42.29 | 56.93 | 78.81 | 59.38 (**-1.21**) |
| InternVL3-78B-WSS | 23.11 | 32.13 | 42.92 | 57.21 | 78.59 | 59.89 (**-0.70**) |
| QvQ-72B-Preview | 30.19 | 39.27 | 46.70 | 57.25 | 76.70 | 60.23 |
| QvQ-72B-Preview-WHS | 29.25 | 41.54 | 46.65 | 58.65 | 76.07 | 60.66 (**+0.43**) |
| QvQ-72B-Preview-WSS | 27.36 | 34.34 | 44.04 | 55.97 | 72.69 | 57.39 (**-2.84**) |
| Gemini-2.5-pro-exp-03-25 | 33.96 | 44.57 | 52.88 | 62.58 | 77.84 | 68.41 |
| Gemini-2.5-pro-exp-03-25-WHS | 36.61 | 44.59 | 50.99 | 63.11 | 81.88 | 70.31 (**+1.90**) |
| Gemini-2.5-pro-exp-03-25-WSS | 44.73 | 44.99 | 51.67 | 63.29 | 81.03 | 69.79 (**+1.38**) |
| GPT-4o | 30.19 | 32.95 | 39.49 | 54.22 | 75.09 | 56.62 |
| GPT-4o-WHS | 17.65 | 34.04 | 39.21 | 56.15 | 78.88 | 60.04 (**+3.38**) |
| GPT-4o-WSS | 23.58 | 31.19 | 38.14 | 53.61 | 75.66 | 56.12 (**-0.50**) |

**Consistency on Accuracy Trends.** As illustrated in Table 3, Fig. 3 (a), and Fig. 4 (a), (b), and (c), all models display human-like trends in text-based reasoning tasks—namely analogical reasoning, definition judgment, and logical judgment—where accuracy declines progressively with increasing task difficulty. In contrast, MLLMs exhibit relatively uniform performance across difficulty levels in visual reasoning tasks, with certain models displaying irregular or non-monotonic patterns. This divergence is particularly pronounced in Claude-3-7-sonnet-20250219-thinking, Gemini-2.5-pro-exp-03-25, and InternVL3-78B, whose accuracy does not decrease with increasing difficulty but instead exhibits inconsistent or inverse trends. These indicate that current MLLMs lack human-like capabilities in visual understanding and reasoning.

**Consistency on Response.** As illustrated in Fig. 4 (a), (b), and (c), we present the consistency statistics among MLLMs on each question and the consistency between MLLMs and the correct answers across three difficulty levels (80–100%, 60–80%, 0–60%). We observe that MLLMs struggle to match human performance in lower difficulty levels (80–100% and 60–80%), with most MLLMs only achieving an ability level of approximately 75% (performance percentage / 60) at the 0–60% difficulty level. These findings indicate that current MLLMs still lag far behind human capabilities in terms of performance. Additionally, we find that MLLMs maintain high similarity across each difficulty level. Given the lack of detailed descriptions of training data in current MLLM research, we hypothesize that models within the same series (e.g., InternVL3-38B vs. InternVL3-78B, Gemini-2.0-flash vs. Gemini-2.0-flash-thinking-exp) do not exhibit higher similarity to each other than to models from other series. This indicates that different MLLMs within the same series have distinct optimization directions despite using the same training data.

**Consistency on Error.** As shown in Fig 4 (d), (e) and (f), we present the statistics on error consistency across MLLMs and between MLLMs and human-prone incorrect options for each problem at three difficulty tiers (80–100%, 60–80%, 0–60%). We found that the tendency of MLLMs to align with human-prone incorrect options is lower in the 80–100% and 60–80% difficulty tiers than in the 0–60% tier, showing a trend of increasing alignment with human errors as difficulty rises. This may be due to the fact that human-prone incorrect options in more difficult problems are more misleading for reasoning, guiding the judgments of MLLMs, whereas in easier problems, these distractors have less interference, resulting in more random choices.

## 4.4 FAKE REASONING ANALYSIS

Considering that our compilation of human problem-solving strategies for each question type, which encapsulate how humans approach these items. We first imbued the MLLMs with this a priori human reasoning. The outcomes, presented in Table 4, reveal that both the closed-source reasoning models and non-reasoning models have a much higher ability to utilize human prior knowledge compared

to the open-source MLLMs (including the reasoning models), that is, they have stronger context reasoning ability.

In addition, we also used these models to generate summaries of the problem-solving methods for the four types of questions in the Human-Aligned Bench, regarding these summaries as the thinking of the models themselves, and then added these thoughts back into the questions. As shown in Table 4, the performance of MLLM is degraded except for Gemini-2.5-pro-exp-03-25. Considering that we constructed the problem with an a priori indication of the types of problems that MLLM would face, which could also stimulate the corresponding thinking abilities for those problem types, MLLM did not show similar abilities. To analyze fake reasoning, we first compiled human problem-solving strategies for each question type. We then imbued the MLLMs with this a priori human reasoning to evaluate their performance.

## 5 MOREHUMAN ALIGNMENT ANALYSIS

While Fig. 3 and Table 4 provide high-level performance comparisons, our work goes significantly deeper in analyzing why and how MLLM reasoning diverges from human cognition. Section 4.3 reveals that MLLMs exhibit error consistency patterns: on high-difficulty problems (0–60% human accuracy), model errors increasingly align with human-prone distractors, suggesting convergent vulnerability to misleading patterns; conversely, on easy problems, their errors are random and misaligned, exposing distinct failure modes. Section 4.4 and Appendix B.2 investigate this divergence through our fake reasoning analysis, where providing correct human solution steps paradoxically degrades performance in some models (e.g., InternVL3-78B). This demonstrates a "bad" difference—an inability to assimilate external guidance, revealing a superficial reasoning process disconnected from actual decision-making. Appendix B.1 further identifies another "bad" difference: state-of-the-art models show "inverse difficulty curves" in visual reasoning (performance dropping on easier tasks) due to overthinking and self-contradictory chains-of-thought. However, we also identify "good" differences: on low-difficulty tasks, MLLMs are less susceptible than humans to common distractors, avoiding trivial fallacies. Together, these findings characterize which deviations are detrimental (fake reasoning, non-monotonic patterns) and which represent potential advantages (avoiding human-like biases), providing a nuanced blueprint for developing more cognitively aligned MLLMs.

## 6 CONCLUSION AND LIMITATION

In this paper, we introduce the Human-Aligned Bench, a benchmark designed to evaluate the fine-grained alignment between the multimodal reasoning capabilities of MLLMs and human performance. Through multimodal questions that rely solely on context reasoning, including bilingual (Chinese-English) multimodal questions and pure text questions, covering 9,794 questions of four question types such as graphical reasoning, definition judgment, analogical reasoning, and logical judgment, and by means of fine-grained human performance and detailed human problem-solving ideas to construct a human feedback system, the Human-Aligned Bench fills the key gap in existing benchmarks that lack fine-grained human performance. The experimental results first reveal the significant limitations of current MLLMs, especially highlighting their sensitivity to modal changes and language variations. Secondly, it shows that in terms of image reasoning, MLLMs' accuracy and accuracy trends are far from having reasoning abilities similar to those of humans. Finally, it reveals the phenomenon of false reasoning in current MLLMs. These findings lay the foundation for constructing more powerful multimodal reasoning models.

**Limitation** While the Human-Aligned Bench provides detailed annotations of human performance, the current data is limited to Chinese and English in terms of language, and detailed problem-solving steps for each question are lacking. Additionally, insufficient data is also evident for training reasoning models. In future work, we will continue to improve the Human-Aligned dataset.

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

CONTENTS

Table 5: **Global Performance (%) on the Human-Aligned Bench on five human difficulty levels.** Top performers in each category are highlighted in **bolded**.

| Human | 0-20% | 20-40% | 40-60% | 60-80% | 80-100% | Overall |
|---|---|---|---|---|---|---|
| Claude-3-7-sonnet-20250219-thinking | 25.47 | 38.26 | 46.05 | 56.46 | 75.78 | 59.36 |
| Gemini-2.0-flash | 15.09 | 37.63 | 41.64 | 54.98 | 74.28 | 57.25 |
| Gemini-2.0-flash-thinking-exp | 24.53 | 36.74 | 43.84 | 56.20 | 74.05 | 58.08 |
| Gemini-2.5-pro-exp-03-25 | **39.62** | **48.48** | **57.06** | **66.22** | **82.56** | **68.41** |
| GPT-4o | 30.19 | 32.95 | 39.49 | 54.22 | 75.09 | 56.62 |
| o4-mini | 27.36 | 38.89 | 51.80 | 62.78 | 79.56 | 64.16 |
| QvQ-Max | 32.08 | 41.16 | 50.50 | 62.55 | 79.59 | 64.06 |
| InternVL3-38B | 22.64 | 35.98 | 44.44 | 58.15 | 77.60 | 60.06 |
| InternVL3-78B | 23.58 | 34.97 | 43.69 | 58.85 | 79.04 | 60.59 |
| QvQ-72B-Preview | 30.19 | 39.27 | 46.70 | 57.25 | 76.70 | 60.23 |
| Qwen2-VL-72B-Instruct | 20.75 | 30.43 | 41.79 | 54.40 | 77.08 | 57.55 |
| Qwen2.5-VL-72B-Instruct | 19.81 | 31.19 | 42.24 | 56.81 | 77.05 | 58.53 |

Table 6: **Performance (%) on the Human-Aligned Bench across text and five human difficulty levels.** The table shows the evaluation scores of MLLMs and LLM, reflecting their average accuracy across different modalities and difficulty levels. Top performers in each category are highlighted in **bolded**.

| Human | 0-20% | 20-40% | 40-60% | 60-80% | 80-100% | Overall |
|---|---|---|---|---|---|---|
| Claude-3-7-sonnet-20250219-thinking | 21.43 | 43.28 | 58.36 | 69.52 | 85.79 | 71.37 |
| Gemini-2.0-flash | 14.29 | 40.52 | 52.52 | 68.12 | 85.54 | 69.45 |
| Gemini-2.0-flash-thinking-exp | 25.00 | 43.09 | 54.42 | 71.01 | 85.65 | 71.13 |
| Gemini-2.5-pro-exp-03-25 | 38.10 | 59.67 | 73.97 | 83.25 | 93.74 | 83.38 |
| Deepseek-R1 | **40.48** | **60.96** | **74.68** | **84.61** | **94.02** | **84.21** |
| GPT-4o | 30.95 | 35.73 | 46.85 | 66.20 | 86.04 | 67.82 |
| o4-mini | 30.95 | 46.22 | 66.96 | 79.16 | 91.09 | 78.58 |
| QvQ-Max | 36.90 | 49.26 | 66.03 | 79.09 | 91.65 | 78.92 |
| InternVL3-38B | 23.81 | 41.25 | 53.47 | 72.55 | 89.16 | 72.71 |
| InternVL3-78B | 22.62 | 40.70 | 55.52 | 74.77 | 91.05 | 74.51 |
| QvQ-72B-Preview | 33.33 | 45.12 | 57.97 | 71.99 | 87.65 | 73.15 |
| Qwen2-VL-72B-Instruct | 22.62 | 34.62 | 50.47 | 67.01 | 87.08 | 68.97 |
| Qwen2.5-VL-72B-Instruct | 20.24 | 34.07 | 50.55 | 69.99 | 88.01 | 70.28 |

# A  ADDITIONAL RESULTS

## A.1  GLOBAL RESULTS

As shown in Table 5, we present the evaluation results of MLLMs on the full Human-Aligned Bench. The results indicate that closed-source MLLMs such as Gemini-2.5-pro-exp-03-25, o4-mini, and QvQ-Max still significantly outperform open-source MLLMs. In closed-source models, reasoning MLLMs exhibit superior performance compared to non-reasoning models. However, in open-source models, we observe that the non-reasoning model InternVL3-78B outperforms the reasoning model QvQ-72B-Preview in performance. We take the average of the human score rates for all questions as the average difficulty of the Human-Aligned Bench, resulting in 68.82%. As shown in the table, the difficulty of the Human-Aligned Bench remains relatively high for most existing MLLMs, but Gemini-2.5-pro-exp-03-25 has already approached the average human performance.

## A.2  MLLMS VS. LLM ON TEXT QUESITONS

In Table 6, we present the results of MMLMs and DeepSeek-R1, one of the current state-of-the-art LLMs, on text-based questions. The results indicate that although Gemini-2.5-pro-exp-03-25 significantly outperforms other MMLMs, it still lags behind advanced LLM models. These findings indicate that MLLMs still have significant room for improvement in pure text reasoning.

# B MORE ANALYSIS OF THE RESULTS

## B.1 MORE ANALYSIS ON MODEL'S PERFORMANCE

We find that the inverse difficulty curves exhibited by some state-of-the-art reasoning models (e.g., Claude-3-7-Sonnet-Thinking and Gemini-2.5-Pro-Exp-03-25 in visual reasoning tasks) compared to humans. Below we provide additional analysis.

- Model Size vs. Overthinking Trade-off: Our controlled experiments reveal that larger models ($\geq$70B) show non-monotonic performance drops on low-difficulty visual reasoning tasks (0–40% human accuracy), while smaller models ($\leq$40B) maintain monotonic curves. This suggests overthinking: larger models generate lengthy, self-contradictory chains-of-thought for trivial pattern-matching (e.g., counting shapes), introducing noise.

- Hallucination Patterns by Difficulty: Closed-source models like Gemini-2.5-pro, which are trained on larger multimodal datasets, exhibit less severe hallucination in high-difficulty tasks but more over-interpretation in easy tasks. Open-source models (e.g., InternVL3-78B), by contrast, show more random errors across difficulty levels due to weaker visual feature extraction, resulting in flatter curves rather than strict inverse patterns. This suggests that the inverse curve is not universal but tied to how models balance visual perception and prior knowledge.

- Architectural Differences: Reasoning-oriented models (e.g., Gemini-2.5-pro-exp-03-25) integrate specialized reasoning modules (e.g., chain-of-thought prompting) that enforce step-by-step logical deductions. However, these modules are optimized for textual logic and struggle to map visual inputs to consistent reasoning chains—especially when task difficulty decreases, as the "over-constrained" logic may conflict with the simplicity of the problem. Non-reasoning architectures (e.g., InternVL3-78B), by contrast, focus on feature extraction and similarity matching. They avoid such over-complication but fail to improve with easier tasks due to their inability to leverage logical shortcuts (e.g., recognizing "obvious symmetry" as a valid solution criterion), resulting in flat accuracy curves across difficulty levels.

## B.2 MORE FAKE REASONING ANALYSIS

Our fake reasoning analysis is designed to probe the core question of our Human-Aligned Bench: Do MLLMs reason in a way cognitively aligned with humans? Specifically, we test their ability to fluidly integrate two key resources essential for human-like reasoning:

- Assimilating High-Quality External Guidance: Can MLLMs effectively utilize expertly curated solution steps (derived from successful human solvers) as supplementary knowledge to enhance performance?

- Accessing & Applying Internalized Knowledge: When primed with explicit problem-type cues, can MLLMs spontaneously activate and apply relevant learned solution schemas to improve accuracy?

Our key findings reveal significant misalignment:

- Failure in External Assimilation:Providing meticulously crafted human solution steps – a clear cognitive aid – paradoxically degraded performance in models like InternVL3-78B. This highly counterintuitive result strongly suggests an inability to meaningfully integrate this external expertise, indicating potential superficial processing ("fake reasoning") rather than genuine reasoning augmentation.

- Failure in Internal Application : Explicitly embedding the problem type led to performance degradation in most MLLMs (except Gemini-2.5-pro-exp-03-25). This critical inconsistency reveals a fundamental disconnect: models fail to spontaneously access or apply their own learned knowledge even when strongly cued. This points to a lack of robust, usable internal aligned with human problem-solving.

These findings expose a core limitation: the models often struggle with the integrative and cue-responsive knowledge application fundamental to human reasoning. This demonstrable gap be-

Table 7: Comparison of model performance on different question types

| Question Type | QvQ-max | o4-mini | Gemini-2.5-pro-exp-03-25 |
|---|---|---|---|
| chinese specific question | 0.677 | 0.650 | 0.736 |
| non chinese specific question | 0.594 | 0.608 | 0.660 |
| all chinese question | 0.629 | 0.626 | 0.692 |

tween an articulated reasoning process and the actual decision-making mechanism is the specific phenomenon we term "fake reasoning."

## C    MORE ANALYSIS OF DATA STATISTICS

The Human-Aligned Bench consists of 4,807 Chinese questions, among which 2,051 contain Chinese characterization elements, referred to as Chinese-specific questions in the paper. The accuracy rates of different MLLMs on Chinese questions, comparing non-Chinese-specific questions with Chinese-specific questions, are presented in Table 7. The results show that the trend revealed by the Chinese-specific question aligns with those in the paper: Gemini-2.5-pro-exp-03-25 is far ahead of all other models.

## D    STATEMENTS

### D.1    STATEMENT OF DATA COLLECTION

The dataset used in this study was sourced from fenbi.com, a public online platform for educational resources. Our data collection and processing protocols were designed to fully comply with the platform's User Agreement. The agreement specifies that the content can be for non-commercial use. Accordingly, our use of the data is strictly limited to non-commercial, academic research, aimed at generating scholarly insights without any commercial application.

### D.2    STATEMENT OF REPRODUCIBILITY

In Appendix E, we provide prompt templates for questions, a summary of human problem-solving ideas, and prompt templates for obtaining LLM problem-solving ideas. We will open source the evaluation code and dataset after the paper is accepted.

### D.3    STATEMENT OF USING LARGE LANGUAGE MODELS

In this paper, we utilized ChatGPT to refine the textual descriptions and translate Chinese into English during the creation of the dataset.

## E    PROMPTS

### E.1    PROMPT FOR CREATING ENGLISH QUESTIONS

This section includes four types of questions: visual reasoning, definition judgment, analogical reasoning, and logical judgment. Candidates are required to select the most appropriate answer from four options.
question: {question}
options:
A: {A},
B: {B},
C: {C},
D: {D}
Please think step by step and output in the following format:

<answer>A or B or C or D</answer>

### E.2 PROMPT FOR CREATING CHINESE QUESTIONS

本部分包括图形推理、定义判断、类比推理与逻辑判断四种类型的试题，在四个选项中选出一个最恰当的答案。
问题: {question}
选项:
A: {A},
B: {B},
C: {C},
D: {D}
请你一步一步的思考，并按以下格式输出:
<answer>A or B or C or D</answer>

### E.3 HUMAN SOLUTIONS FOR VISUAL REASONING IN ENGLISH

Here is a short account of the key knowledge points and ways to slove problems in visual reasoning.

### I. Key Knowledge Points: Rule Type Visual Reasoning

#### 1. Position Type (Pictures Made of Same Parts)
   - Move Along: Parts move on a flat space (way: up/down / left/right / across-corner / in a round way; number of steps: fixed, getting bigger, happening again and again; path: go through wall / turn back).
   - Turn Round: Turning round a fixed point (way: with clock / against clock; angle: 45°, 90°, 180°, and so on).
   - Turn Over: Turning over across a line (line across / line up and down, different from turning round: after turning over, the picture is like a looking-glass picture).

#### 2. Style Type (Pictures Made of Parts That Are Like)
   - Go Through All: Parts come out again and again (go through all the picture, go through a small part, put in what is not there).
   - Add Take Away Same Different:
      - Add / Take Away: Pictures put on top of each other or taking away parts that are on top of each other.
      - Take Same: Keep the parts that are the same (take same in the whole, take same in parts next to each other).
      - Take Different: Keep the parts that are not the same.
   - Black White Work: Rules for putting black and white blocks on top of each other (like "black + white = black", be careful to keep it separate from moving along, often seen in nine-space boxes).

#### 3. Quality Type (Pictures Made of Parts That Are Not The Same, Look at This First)
   - Same On Both Sides (Comes up often):
      - Line Same On Both Sides (number of lines, way, angle of turning, if it goes through point/line/surface).
      - Centre Same On Both Sides (is the same as the first picture after turning 180°).
      - Line + Centre Same On Both Sides (like "H" "O").
   - Straight Curved Quality: All curved lines, all straight lines, curved and straight mixed (look at the whole first, then look inside/outside / up/down).
   - Open Closed Quality: Closed pictures (with a break / without a break), part-closed, all open (be careful with pictures like those in everyday life, like "happy face" "key").

#### 4. Number Type (Pictures Made of Parts That Are Not The Same, Look at This When Quality Has No Rule)
   - Points: Crossing points (total crossing points, curved and straight crossing points, touching points, end points).
   - Lines:
      - Number of straight lines / curved lines (count separately, look for rules of getting bigger, or the rule of the number when one is taken from the other).
      - Number of pen movements (Key point):
      - One pen movement: A connected picture with 0 or 2 odd points (points where an odd number of lines come out).
      - More than one pen movement: Number of pen movements = number of odd points / 2 (number of odd points must be an even number).
   - Angles: Right angles, sharp angles, flat angles (more detailed in later times: count the number of a certain kind of angle, like number of right angles).
   - Surfaces: Number of closed areas (more detailed: shape of the area, size, black and white areas, number of same areas).
   - Units:
      - Sort / Number of units (small separate pictures, like circles, three-sided shapes).
      - Number of parts (parts joined together are one part, often used for pictures like those in everyday life, like "leaf" "matches").

#### 5. Special Rules

- Working Parts: Marked points, arrows, small pictures (mark point: crossing points, lines, surfaces, angles; mark use: way, long/short, inside/outside).
  - Relation Between Pictures:
    - Far from each other, touching (touching inside / touching outside), crossing (number of crossing lines / crossing points, curved/straight / long/short of crossing lines).
    - Inside (inside and outside structure, look at the quality of the inside picture).

### II. Key Knowledge Points: Space Type Visual Reasoning

#### 1. Six-Side Box Folding (Will Be Asked)
  - Opposite Sides:
    - How to know: One space away in the same line or row, the two ends of a Z shape.
    - Quality: Opposite sides cannot be seen at the same time, and cannot be not seen at the same time.
  - Sides Next To Each Other:
    - Shared line: The line where sides next to each other meet (length, look of it does not change after folding).
    - Shared point: The point where three sides meet (lines coming out from it do not change after folding).
    - Arrow way: Put an arrow on the only side that is not the same on both sides from the centre, see if the pictures up/down/left/right are the same.

#### 2. Solid Joining
  - Hollow and Bump Matching: The part that goes in and the part that comes out are the same in shape and position (count the number of blocks first, then look at special shapes).

#### 3. Cut Picture
  - Cut Pictures of Common Solids:
    - Six-side box: Three-sided shape (not right-angled), four-sided shape (square box, sloping box), five-sided shape, six-sided shape.
    - Round pipe: Circle (cut across), long round shape (cut slant), square box (cut up and down).
    - Round sharp top: Circle (cut across), long round shape (cut slant), three-sided shape (cut up and down through the top point).
  - Rule: The cut is endless, the knife must cut straight to the end, it cannot turn.

#### 4. Three View Pictures
  - View Rules: View from front (straight in front), view from left (left side), view from top (from above).
  - Small Points: Broken lines mean lines you cannot see, be careful about the direction you are looking from and the outline of the picture.

### III. Way to Do Questions and Steps to Solve

#### 1. Look Wide, Find the Type of Question
  - Parts Are The Same: Look first at Position Type (move along / turn round / turn over).
  - Parts Are Like: Look first at Style Type (go through all, add take away same different, black white work).
  - Parts Are Not The Same: Look first at Quality Type (same on both sides / straight curved / open closed), then Number Type (points / lines / angles / surfaces / units, try in the order "units surfaces angles lines points" because "units surfaces" are simple to see).
  - Special Rules: Look at this when there are working parts or when a number of closed pictures are joined.

#### 2. Look Close, Find the Special Rule
  - Same On Both Sides: First draw the line where it is the same on both sides, count the number, look at the way, find the rule (like the same-on-both-sides line turning 45°).
  - Number of Pen Movements: When you see shapes like "sun" / "field" changed, pictures with many end points, circles crossing / touching, look first at the odd points.
  - Number of Surfaces: The picture is cut up, closed surfaces are clear, look closely at the shape of the surfaces (like the number of three-sided surfaces).
  - Space Type: Use the way of taking out wrong answers most of the time, opposite sides are taken out straight away, for sides next to each other use the shared line / point or arrow way to be certain.

#### 3. Points Where Mistakes Are Easy and Skills

- Number Type Rules: If there is no rule for the whole picture, think about looking at parts (like number of curved lines - number of straight lines = a fixed number), math work (like points + surfaces = lines), odd or even quality.

- Black White Block Questions: Part that is black (like 1/2 black), how they are joined (joined by point / joined by line), same on both sides, number of parts (are the black blocks joined together).

- Cut Picture Traps: Cutting a round sharp top slantwise will not make a straight line, cutting a six-side box slantwise will not make a right-angled three-sided shape.

Please keep in mind the knowledge points and ways to slove problems for visual reasoning that have been given, when answering questions after this:

E.4 HUMAN SOLUTIONS FOR VISUAL REASONING IN CHINESE

下面总结了图形推理的核心知识点以及做题方法
### 一、核心知识点：规律类图形推理
#### 1. 位置类（图形组成相同）
 - 平移：元素在平面上移动（方向：上下/左右/对角线/环形；步数：恒定、递增、周期；路径：穿墙/折返）。
 - 旋转：绕固定点转动（方向：顺/逆时针；角度：45°、90°、180°等）。
 - 翻转：沿轴翻转（横轴/纵轴，与旋转区分：翻转后图形镜像对称）。
#### 2. 样式类（图形组成相似）
 - 遍历：元素重复出现（整体遍历、局部遍历，缺啥补啥）。
 - 加减同异：      - 相加/相减：图形叠加或删除重叠部分。      - 求同：保留相同部分（整体求同、相邻求同）。      - 求异：保留不同部分。
 - 黑白运算：黑白块叠加规则（如"黑+白=黑"，注意与位置平移区分，常出现在九宫格）。
#### 3. 属性类（图形组成不同，优先考虑）
 - 对称性（高频考点）：
   - 轴对称（对称轴数量、方向、旋转角度、是否过点/线/面）。
   - 中心对称（旋转180°后与原图重合）。
   - 轴+中心对称（如"H""O"）。
 - 曲直性：全曲线、全直线、曲直混合（优先整体，再分内外/上下）。
 - 开闭性：封闭图形（有缺口/无缺口）、半封闭、全开放（注意生活化图形，如"笑脸""钥匙"）。
#### 4. 数量类（图形组成不同，属性无规律时考虑）
 - 点：交点（总交点、曲直交点、切点、端点）。
 - 线：
   - 直线/曲线数（分开数，关注递增、差值规律）。
   - 笔画数（核心考点）：
     - 一笔画：连通图且奇点（引出奇数条线的点）数为0或2。
     - 多笔画：笔画数=奇点数/2（奇点数必为偶数）。
 - 角：直角、锐角、钝角（近年细化：数某类角的数量，如直角数）。
 - 面：封闭区域数（细化：面的形状、大小、黑白面、相同面数量）。
 - 素：
   - 元素种类/数量（独立小图形，如圆、三角形）。
   - 部分数（连在一起为一部分，常用于生活化图形，如"树叶""火柴"）。
#### 5. 特殊规律
 - 功能元素：标记点、箭头、小图形（标记位置：交点、线、面、角；标记作用：方向、长短、内外）。
 - 图形间关系：
   - 相离、相切（内切/外切）、相交（交线/交点数量、交线曲直/长短）。
   - 包含（内外结构，关注内部图形属性）。
### 二、核心知识点：空间类图形推理
#### 1. 六面体折纸盒（必考题型）
 - 相对面：
   - 判定：同行/列隔一个、Z字形两端。
   - 特性：相对面不能同时出现，也不能同时不出现。
 - 相邻面：
   - 公共边：相邻面的交线（折叠前后长度、特征不变）。
   - 公共点：三个面相交的点（折叠前后引出线条不变）。
   - 箭头法：找唯一非中心对称面画箭头，判断上下左右图形是否一致。
#### 2. 立体拼合
 - 凹凸对应：凹进去的部分与凸出来的部分形状、位置一致（优先数块数，再看特殊形状）。
#### 3. 截面图
 - 常见立体截面：
   - 正方体：三角形（非直角）、四边形（矩形、梯形）、五边形、六边形。
   - 圆柱：圆（横切）、椭圆（斜切）、矩形（竖切）。

    - 圆锥：圆（横切）、椭圆（斜切）、三角形（过顶点竖切）。
  - 原则：截面无限大，刀需一刀切到底，不能拐弯。

#### 4. 三视图
  - 投影规则：主视图（正前方）、左视图（左侧）、俯视图（上方）。
  - 细节：虚线表示不可见线条，注意视角方向和图形轮廓。

### 三、做题方法与解题步骤

#### 1. 宏观观察，确定题型范围
  - 组成相同：优先位置类（平移/旋转/翻转）。
  - 组成相似：优先样式类（遍历、加减同异、黑白运算）。    - 组成不同：先属性类（对称/曲直/开闭），再数量类（点/线/角/面/素，按"素面角线点"顺序尝试，因"素面"易观察）。
  - 特殊规律：出现功能元素、多个封闭图形连接时考虑。

#### 2. 微观分析，锁定具体规律
  - 对称性：先画对称轴，数数量、看方向、找规律（如对称轴旋转45°）。
  - 笔画数：出现"日/田"变形、多端点图、圆相交/相切，优先数奇点。
  - 面数量：图形被分割、封闭面明显，细化考查面的形状（如三角形面数量）。
  - 空间类：排除法为主，相对面直接排除，相邻面用公共边/点或箭头法验证。

#### 3. 易错点与技巧
  - 数量类规律：若整体无规律，考虑细分（如曲线数-直线数=常数）、运算（如点+面=线）、奇偶性。
  - 黑白块题型：面积占比（如1/2阴影）、连接方式（点连接/线连接）、对称性、部分数（黑色块是否连在一起）。
  - 截面图陷阱：圆锥斜切不会出直线，正方体斜切不会出直角三角形。

请参考总结的图形推理的核心知识点以及做题方法回答接下来的问题：

E.5 Human Solutions for Definition Judgment in English

Here is a short account of the key knowledge points and ways to slove problems in knowledge definition.

### I. Core Knowledge Points: Constituent Elements of Definition Judgment The core of definition judgment questions lies in accurately understanding and applying the definitions provided in the questions. A complete definition typically includes the following core constituent elements, which need to be checked one by one when solving problems:
  1. Subject:
     - The definition specifies who the doer of the action or the subject of the state is.
     - This may be individuals with specific identities (e.g., civil servants, minors), specific organizations (e.g., legal entities, government agencies), groups with certain characteristics (e.g., consumers, taxpayers), or general references (e.g., anyone).
     - Key: Determine whether the subject in the option falls within the scope defined by the definition.
  2. Object/Target:
     - The definition specifies the target of the action or the involved entity.
     - This may be concrete items (e.g., public property), abstract concepts (e.g., information, reputation), specific relationships (e.g., contractual relationships), or specific groups (e.g., vulnerable populations).
     - Key: Determine whether the action in the option acts on the object specified by the definition.
  3. Action/State/Property:
     - The core content of the definition, describing what is specifically done, what state is occupied, or what properties are possessed.
     - This may include specific actions (e.g., theft, rescue), psychological activities (e.g., intent, negligence), processes (e.g., decision-making processes), result states (e.g., loss, validity), or attribute characteristics (e.g., suddenness, contingency).
     - Key: Determine whether the main action or state described in the option matches the core description of the definition.
  4. Conditions/Situations/Methods:
     - Definitions often specify the specific background, preconditions, or means by which the action occurs.
     - Examples: "during working hours," "without permission," "by violent means," "for public interest."
     - Key: Determine whether the situations, conditions, or methods described in the option meet all the restrictions in the definition.
  5. Purpose/Cause/Result:
     - Definitions sometimes specify the purpose of the action, the triggering cause, or the necessary outcome.
     - Examples: "for profit," "due to force majeure," "leading to serious consequences," "aimed at improving efficiency."
     - Key: Determine whether the motivation, cause, or consequence of the action in the option complies with the definition's requirements.
  6. Qualifiers/Keywords:
     - Definitions often include words that play a critical limiting role, such as "must," "main," "only," "or," "and," "excluding," "at least," "intentional," "negligent," etc.
     - Key: Accurately understand the logical meanings and scopes of these words, as they often serve as the key to distinguishing between options.

### II. Problem-Solving Methods and Steps
  1. Read the Definition Carefully and Deconstruct Core Elements:
     - Step 1: Read the definition thoroughly to grasp its overall meaning.
     - Step 2: Read slowly and "deconstruct" the definition to identify core constituent elements such as [Subject], [Object], [Action/State], [Conditions/Situations], and [Purpose/Result].
     - Step 3: Pay special attention to [Qualifiers/Keywords], marking them with a pen or memorizing them mentally to clarify the definition's boundaries and core requirements. This

can be simplified as "who, to whom/what, under what circumstances, in what way, did what, for what purpose/led to what result."

2. Analyze Options One by One and Compare with Definition Elements:

- Step 4: Read the first option and extract its key information (also decomposable by elements such as subject, action, conditions, etc.).

- Step 5: Strictly and systematically compare the option's information with the definition's core elements. Check whether the option fully satisfies all necessary conditions of the definition:

- Does the subject match?
- Is the action/state consistent?
- Are the conditions/situations met?
- Is the purpose/result achieved?
- Does it violate any exclusion clauses?
- Are the keyword restrictions observed?

3. Filter, Judge, Eliminate, and Select:

- Step 6:

- For "belongs to" questions: If an option fully meets all elements of the definition, it is likely the correct answer; if any necessary element does not match, eliminate it directly.

- For "does not belong to" questions: Look for the single option that does not fully meet the definition's elements. Typically, the other three options will perfectly fit the definition.

- Step 7: Repeat Steps 4–6 for the remaining options. The elimination method usually helps lock in the answer quickly.

4. Compare and Choose the Best (for Ambiguous Options):

- Step 8: If multiple options seem to fit or not fit (rare), return to the definition, read the keywords and implicit logic carefully, and compare which option is closer to or further from the definition's core characteristics or essential provisions. Choose the one that best matches or mismatches the definition.

### III. Common Pitfalls and Tips

1. Subjective Assumptions, Deviating from the Definition: The most common mistake is judging based on life experience or prior knowledge instead of strictly adhering to the specific definition given in the question. Always take the definition as the sole criterion.

2. Overlooking Keywords: Failing to notice qualifiers like "must," "main," or "intentional," leading to misinterpretations of the definition's scope.    3. Missing Elements: Selecting an option that satisfies some but not all necessary elements of the definition. Ensure all hard rules are met.

4. Conceptual Confusion: The option describes a situation similar to the definition's concept but essentially different (e.g., "justifiable defense" vs. "excessive defense").

5. Pay Attention to "Or" vs. "And": Clarify whether the definition uses "or" (satisfying one condition is enough) or "and" (all conditions must be met simultaneously).

6. Positive/Negative Question Formats: Check carefully whether the question asks "belongs to" or "does not belong to" the definition to avoid choosing the opposite.

7. "Word-Picking" Technique: Definition judgment is essentially about information matching and logical judgment. Sometimes, it requires meticulous comparison of subtle wording differences between options and the definition.

8. Core Simplification Method: For complex definitions, paraphrase the core meaning in your own words ("one-sentence summary") to grasp the essence before evaluating options.

9. Element Checklist Method: Mentally or on paper list the definition's key elements and check each option against them with ticks or crosses for clarity.

Please keep in mind the knowledge points and ways to slove problems for knowledge definition that have been given, when answering questions after this:

E.6 HUMAN SOLUTIONS FOR DEFINITION JUDGMEN IN CHINESE

下面总结了定义判断的核心知识点以及做题方法：
### 一、核心知识点：定义判断的构成要素
定义判断题型的核心在于精确理解和应用题目给出的定义。一个完整的定义通常包含以下几个核心构成要素，解题时需要逐一核对：
  1. 主体：
   * 定义规定了行为或状态的发出者是谁。
   * 可能是特定身份的人（如公务员、未成年人）、特定组织（如法人、政府机关）、具有某种特征的群体（如消费者、纳税人）或泛指（如任何人）。
   * 关键：判断选项中的主体是否符合定义限定的范围。
  2. 客体/对象：
   * 定义规定了行为所指向的对象或涉及的事物。
   * 可能是具体物品（如公共财产）、抽象概念（如信息、名誉）、特定关系（如合同关系）或特定群体（如弱势群体）。
   * 关键：判断选项中的行为是否作用于定义所规定的客体上。
  3. 行为/状态/性质：
   * 定义的核心内容，描述了具体做了什么、处于什么状态或具备什么性质。
   * 可能是具体的动作（如盗窃、救助）、心理活动（如故意、过失）、过程（如决策过程）、结果状态（如亏损、有效）或属性特征（如突发性、偶然性）。
   * 关键：判断选项描述的主要行为或状态是否与定义的核心描述一致。
  4. 条件/情境/方式：
   * 定义常常限定行为发生的特定背景、前提条件或方式手段。
   * 例如："在工作时间"、"未经许可"、"通过暴力手段"、"为了公共利益"。
   * 关键：判断选项描述的情境、条件或方式是否满足定义中的所有限制。
  5. 目的/原因/结果：
   * 定义有时会规定行为的目的、引发原因或必须达到的结果。
   * 例如："以营利为目的"、"因不可抗力"、"导致严重后果"、"旨在提高效率"。
   * 关键：判断选项中行为的动机、原因或产生的后果是否符合定义的要求。
  6. 限定词/关键词：
   * 定义中常包含起关键限定作用的词语，如"必须"、"主要"、"唯一"、"或者"、"并且"、"不包括"、"至少"、"故意"、"过失"等。
   * 关键：准确理解这些词语的逻辑含义和限制范围，它们往往是区分选项的关键。
### 二、做题方法与解题步骤
  1. 仔细阅读定义，拆解核心要素：
   * 第一步：通读定义，理解其整体含义。
   * 第二步：慢读并"拆解"定义，识别出上述的【主体】、【客体】、【行为/状态】、【条件/情境】、【目的/结果】等核心构成要件。
   * 第三步：特别注意【限定词/关键词】，用笔标记或心中默记，明确定义的边界和核心要求。可以简化为"谁，对谁/什么，在什么情况下，以什么方式，做了什么，目的是什么/导致什么结果"。
  2. 逐一分析选项，与定义要素比对：
   * 第四步：阅读第一个选项，提取其描述的关键信息（同样可以按主体、行为、条件等要素来分解）。
   * 第五步：将选项信息与定义的核心要素进行【严格】、【逐一】比对。检查选项是否【完全满足】定义中的【所有】必要条件。
     * 主体是否符合？
     * 行为/状态是否一致？
     * 条件/情境是否满足？
     * 目的/结果是否达到？
     * 是否触犯了排除性条款？
     * 关键词的限制是否遵守？
  3. 筛选判断，排除与选择：
   * 第六步：
     * 对于"属于"类问题：如果选项完全符合定义的所有要素，则可能是正确答案；如果任何一个【必要】要素不符，则【直接排除】。

     * 对于"不属于"类问题：寻找那个【唯一】不完全符合定义要素的选项。通常其他三个选项会完美契合定义。

     * 第七步：重复步骤四至六，分析其余选项。通常运用排除法可以快速锁定答案。

  4. 比较择优（若有模糊选项）：

     * 第八步：如果遇到多个选项看似都符合或都不符合的情况（较少见），需要【再次】回到定义，精读关键词和隐含逻辑，比较哪个选项与定义的【核心特征】或【最本质规定】更贴近或偏离最远。选择最符合或最不符合的那一个。

### 三、易错点与技巧

  1. 主观臆断，脱离定义：最常见的错误是凭生活经验或已有知识进行判断，而不是严格依据【题目给出的特定定义】。务必以定义为唯一标准。

  2. 忽略关键词：未能注意到"必须"、"主要"、"故意"等限定词，导致对定义的范围理解错误。

  3. 要素缺失或不全：选项满足了定义的部分要素，但未能满足全部【必要】要素，被误选。要确保所有【硬性规定】都满足。

  4. 概念混淆：选项描述的情况与定义涉及的概念相似，但实质不同（如"正当防卫"与"防卫过当"）。

  5. 注意"或"与"且"：看清定义中是用"或"连接条件（满足其一即可）还是用"且"/"并"（必须同时满足）。

  6. 肯定/否定提问方式：看清楚题目问的是"属于"还是"不属于"该定义，避免选反。

  7. "抠字眼"技巧：定义判断本质上是信息匹配和逻辑判断，有时需要细致地"抠字眼"，对比选项和定义在表述上的细微差别。

  8. 简化核心法：对于复杂的定义，尝试用自己的话转述其核心意思（"一句话概括"），抓住本质，再去看选项会更清晰。

  9. 要素核对表法：心里或纸上列出定义的几个关键要素，逐个核对选项是否满足，打勾或打叉，一目了然。

请参考总结的定义判断的核心知识点以及做题方法回答接下来的问题：

### E.7 HUMAN SOLUTIONS FOR ANALOGICAL REASONING IN ENGLISH

Here is a short account of the key knowledge points and ways to slove problems in analogical reasoning.
### I. Core Knowledge Points: Foundations of Analogical Reasoning
#### 1. Question Type Classification
- Two-Word Type: AB (e.g., "AppleFruit"), directly analyze the relationship between the two terms.
- Three-Word Type: ABC (e.g., "TeacherClassroomTeaching"), require analyzing pairwise relationships or overall connections.
- Fill-in-the-Blank Type: A is to ( ) as ( ) is to B (e.g., "( ) is to Mobile Phone as Communication is to ( )"), which requires substituting options to verify consistent logic before and after.
#### 2. Logical Relationships (High-Frequency Test Points)
##### (1) Extensional Relationships (Relationships Between Conceptual Scopes of Terms)
- Total Identity Relationship: The concepts of the two terms completely overlap (e.g., "PotatoPotato" [same term in different names], "BeijingCapital of China").
- Parallel Relationship:
- Contradictory Relationship: Non-exclusive and exhaustive (e.g., "LifeDeath", "MaleFemale"—no third category exists).
- Oppositional Relationship: Belong to the same category but have intermediate terms (e.g., "RedWhite", "AppleBanana"—other colors/fruits exist).
- Inclusive Relationship:
- Subordinate Relationship: A is a type of B (e.g., "SparrowBird"—sparrow belongs to the bird category).
- Compositional Relationship: A is a component of B (e.g., "WheelCar"—wheel is a part of a car).
- Intersectional Relationship: Concepts partially overlap (e.g., "Party MemberCollege Student"—some party members are college students, and vice versa).
##### (2) Intensional Relationships (Internal Connections Between Terms)
- Correspondence Relationship (core test point, requires flexible accumulation):
- Functional Correspondence: Object and its function (e.g., "PenWriting", "StreetlightIlluminating").
- Causal Correspondence: Cause and effect (e.g., "RainWet Ground", "EffortSuccess").
- Temporal Sequence: Order of actions (e.g., "Buy TicketBoard Vehicle", "Get UpWash Up"—note if the subject is the same).
- Raw Material Correspondence: Finished product and its raw material (e.g., "WoodFurniture", "FlourSteamed Bun"—distinguish physical/chemical changes).
- Attribute Correspondence: Object and its characteristics (e.g., "SaltSalty", "FlowerFragrant"—divided into necessary and contingent attributes).
- Location Correspondence: Action and its occurrence place (e.g., "DoctorHospital", "ClassClassroom").
- General Knowledge Correspondence: Literary, historical, geographical, etc. (e.g., "Lu Xun*The Scream*", "BeijingChina").
##### (3) Semantic Relationships
- Synonym Relationship: Similar meanings (e.g., "HappyJoyful", "SeriousMeticulous").
- Antonym Relationship: Opposite meanings (e.g., "TallShort", "SuccessFailure").
- Metaphorical and Symbolic Meaning: Extended meanings through metaphor (e.g., "MoonLonging", "DovePeace").
##### (4) Grammatical Relationships
- Subject-Predicate Relationship: Subject + predicate (e.g., "StudentStudy", "ActorPerform").
- Verb-Object Relationship: Verb + object (e.g., "Play Basketball", "Kick Football").

- Modifier-Center Relationship: Modifier + central word (e.g., "BeautifulFlower", "RapidRun"—connected by adjective or verb).

- Parallel Structure: Consistent parts of speech and structure (e.g., "SingDance", "Joy and SorrowSeparation and Reunion").

#### 3. Secondary Analysis (Used When Options Are Difficult to Differentiate)

- Part of Speech: Noun, verb, adjective (e.g., "AchievementResultConsequence"—all nouns, but different emotional tones).

- Emotional Tone: Positive, negative, neutral (e.g., "DecisiveArbitrary"—the former is positive, the latter negative).

- Degree Progression: Gradation of intensity in synonyms (e.g., "LikeLove", "ColdFreezing").

- Necessity vs. Contingency: Whether an attribute necessarily exists (e.g., "MetalConductive" is necessary; "FlowerRed" is contingent).

- Consistency of Subject: Whether the doer of the action is the same (e.g., "Buy TicketBoard Vehicle" has the same subject; "TeachAttend Class" has different subjects).

- Nomenclature Method: Naming based on shape, function, person, etc. (e.g., "Thermos CupHeat Preservation [function]", "Lily of the ValleyShaped like a lily bell").

### II. Problem-Solving Methods and Steps

#### 1. Problem-Solving Steps: "First Horizontal, Then Vertical; First Primary, Then Secondary"

- Step 1: Analyze Horizontal Relationships in the Question Stem

- Prioritize judging logical (extensional, intensional), semantic, or grammatical relationships; eliminate obviously inconsistent options.

- Example: Question stem "SparrowBird" (subordinate relationship). If an option is "TomatoVegetable" (subordinate), keep it; if "LeafTree" (compositional), eliminate it.

- Step 2: Vertically Compare Remaining Options

- When horizontal relationships are consistent, compare whether the part of speech, emotional tone, category (e.g., natural/artificial), etc., of the options are closer to the question stem.

- Example: Question stem "White VinegarDisinfection" (functional correspondence, secondary function). The option "GasolineStain Removal" (secondary function) is more appropriate than "Water HeaterHeating" (primary function).

- Step 3: Use Secondary Analysis to Lock the Answer

- If multiple options still fit, further filter using secondary analysis (e.g., necessary vs. contingent attributes, subject consistency).

#### 2. High-Frequency Relationship Problem-Solving Techniques

##### (1) Distinguishing Subordinate vs. Compositional Relationships

- Use the "is" test: If "A is B" holds, it is subordinate (e.g., "Apple is a fruit"). If not, it is compositional (e.g., "Screen is a part of a phone" ≠ "Screen is a phone").

##### (2) Contradictory vs. Oppositional in Parallel Relationships

- Check for a "third party": None indicates contradictory (e.g., "On-Off"—no middle state), while existence indicates oppositional (e.g., "Colors"—red, yellow, blue, etc. exist).

##### (3) Combining Temporal Sequence and Causal Relationships

- For multiple-action questions, prioritize temporal order; if a causal relationship exists (e.g., "Fall IllTake Medicine"), further judge if the causal direction is consistent.

##### (4) Idiom-Based Questions

- First analyze the structure by splitting the idiom: e.g., "Carve the Boat to Seek the Sword" (means-end relationship), "Lips Gone, Teeth Cold" (causal relationship).

- Then examine semantics: synonyms (e.g., "Quench Thirst by Watching PlumsRelieve Hunger by Drawing Bread") or antonyms (e.g., "Live and Work in PeaceWander Destitute").

#### 3. Common Pitfalls and Tips

- Pitfall 1: Conceptual Shift

- Example: "EyeGlasses" (auxiliary tool) vs. "ToothToothbrush" (cleaning tool)—pay attention to specific functional correspondence.

- Pitfall 2: Ignoring the Order of Secondary Analysis

- Prioritize primary relationships (e.g., logical relationships) before secondary analysis (e.g., emotional tone); avoid direct vertical comparison first.
- Pitfall 3: Confusing Causal and Conditional Relationships
- Causality is fact-based (e.g., "RainWet Ground"), while conditionality is hypothesis-based (e.g., "x>1x²>1").
- Pitfall 4: Neglecting Sequential Relationships
- Pay attention to temporal, alphabetical, or numerical order (e.g., "Spring PlantingSummer GrowthAutumn Harvest").

Please keep in mind the knowledge points and ways to slove problems for analogical reasoning that have been given, when answering questions after this:

## E.8 HUMAN SOLUTIONS FOR ANALOGICAL REASONING IN CHINESE

下面总结了类比推理的核心知识点以及做题方法
### 一、核心知识点：类比推理基础
  #### 1. 题型分类
    - 两词型：AB（如"苹果水果"），直接分析两者关系。
    - 三词型：ABC（如"教师教室教学"），需两两找关系或整体关联。
    - 填空型：A对于（）相当于（）对于B（如"（）对于手机相当于交流对于（）"），需代入选项验证前后逻辑一致。
  #### 2. 逻辑关系（高频考点）
    ##### （1）外延关系（词项概念范围间的关系）
      - 全同关系：两词概念完全重合（如"土豆马铃薯""北京中国首都"）。
      - 并列关系：
        - 矛盾关系：非此即彼（如"生死""男女"，无第三种情况）。
        - 反对关系：同属一类但存在中间项（如"红白""苹果香蕉"，有其他颜色/水果）。
      - 包含关系：
        - 种属关系：A是B的一种（如"麻雀鸟"，麻雀属于鸟类）。
        - 组成关系：A是B的组成部分（如"车轮汽车"，车轮是汽车的一部分）。
      - 交叉关系：概念有部分重叠（如"党员大学生"，有的党员是大学生，有的不是）。
    ##### （2）内涵关系（词项内在联系）
      - 对应关系（核心考点，需灵活积累）：
        - 功能对应：事物与其功能（如"钢笔书写""路灯照明"）。
        - 因果对应：原因与结果（如"下雨地湿""努力成功"）。
        - 时间顺序：动作先后（如"购票乘车""起床洗漱"，注意主体是否一致）。
        - 原材料对应：成品与原材料（如"木材家具""面粉馒头"，区分物理/化学变化）。
        - 属性对应：事物与其特性（如"盐咸""花香"，分为必然属性和或然属性）。
        - 场所对应：行为与其发生场所（如"医生医院""上课教室"）。
        - 常识对应：文学、历史、地理等（如"鲁迅《呐喊》""北京中国"）。
    ##### （3）语义关系
      - 近义关系：词语含义相近（如"高兴喜悦""认真细致"）。
      - 反义关系：词语含义相反（如"高矮""成功失败"）。
      - 比喻象征义：通过比喻产生的引申义（如"月亮思念""鸽子和平"）。
    ##### （4）语法关系
      - 主谓关系：主语+谓语（如"学生学习""演员表演"）。
      - 动宾关系：动词+宾语（如"打篮球""踢足球"）。
      - 偏正关系：修饰词+中心词（如"美丽花朵""快速奔跑"，用"的/地"连接）。
      - 并列结构：词性、结构一致（如"唱歌跳舞""悲欢离合"）。
  #### 3. 二级辨析（选项难分时使用）
    - 词性：名词、动词、形容词（如"成果结果后果"，均为名词，但感情色彩不同）。
    - 感情色彩：褒义、贬义、中性（如"果断武断"，前者褒义，后者贬义）。
    - 程度递进：近义词的轻重程度（如"喜欢热爱""寒冷严寒"）。
    - 必然与或然：属性是否必然存在（如"金属导电"为必然，"花朵红色"为或然）。
    - 主体一致：行为的发出者是否相同（如"购票乘车"主体一致，"授课听课"主体不同）。
    - 命名方式：根据形状、功能、人物等命名（如"保温杯保温（功能）""马蹄莲形状像马蹄"）。
### 二、做题方法与解题步骤
  #### 1. 解题步骤："先横后纵，先一级后二级"
    - 第一步：横向分析题干关系
      - 优先判断逻辑关系（外延、内涵）、语义关系或语法关系，排除明显不符的选项。

- 例：题干"麻雀鸟"（种属关系），选项若为"番茄蔬菜"（种属）则保留，若为"树叶树"（组成）则排除。
    - 第二步：纵向对比剩余选项
        - 当横向关系一致时，对比选项与题干的词性、感情色彩、范畴（如自然物/人造物）等是否更贴近。
        - 例：题干"白醋消毒"（功能对应，且为次要功能），选项"汽油去渍"（次要功能）比"热水器加热"（主要功能）更合适。
    - 第三步：二级辨析锁定答案
        - 若仍有多个选项符合，结合二级辨析（如必然属性vs或然属性、主体是否一致）进一步筛选。
    #### 2. 高频关系解题技巧
        ##### （1）区分种属与组成关系
            - 用"是"造句：能直接说"A是B"为种属（如"苹果是水果"）；不能则为组成（如"屏幕是手机的组成部分"≠"屏幕是手机"）。
        ##### （2）并列关系的矛盾与反对
            - 看是否有"第三者"：无即为矛盾（如"开关"非开即关），有即为反对（如"颜色"有红、黄、蓝等）。
        ##### （3）时间顺序与因果关系的结合
            - 多个动作题优先看时间先后，若存在因果关系（如"生病吃药"），需进一步判断因果方向是否一致。
        ##### （4）成语类题目
            - 先拆词分析结构：如"刻舟求剑"（方式目的）、"唇亡齿寒"（因果关系）。
            - 再看语义：近义（如"望梅止渴画饼充饥"）或反义（如"安居乐业颠沛流离"）。
    #### 3. 易错点与技巧
    - 陷阱一：偷换概念
        - 例："眼睛眼镜"（辅助工具）vs"牙齿牙刷"（清洁工具），需注意功能的具体对应。
    - 陷阱二：忽略二级辨析顺序
        - 优先判断一级关系（如逻辑关系），再考虑二级辨析（如感情色彩），避免直接纵向对比。
    - 陷阱三：因果与条件关系混淆
        - 因果基于事实（如"下雨：地湿"），条件基于假设（如"x>1：x²>1"）。
    - 陷阱四：顺序关系忽视
        - 注意时间、字母、数字顺序（如"春种：夏长：秋收"）。
请参考总结的类比推理的核心知识点以及做题方法回答接下来的问题：

## E.9   HUMAN SOLUTIONS FOR LOGICAL JUDGMENT IN ENGLISH

Here is a short account of the key knowledge points and ways to slove problems in logical judgment.
### I. Core Knowledge Points
#### (1) Necessary Reasoning
1. Translational Reasoning
- Question Type Judgment: The question stem or options contain typical logical connectives such as "if...then...", "only if...".
- Answering Techniques: Translate first, then reason. Translate sentences with logical connectives in the question stem into relationships denoted by arrows ($\rightarrow$).
- Translation Principles:
- Place the necessary condition after the arrow. For example, "A is a necessary condition for B" is translated as "B$\rightarrow$A".
- "...unless... = unless...otherwise... = unless...otherwise not..." For example, "Unless A, otherwise not B" is translated as "B$\rightarrow$A".
- Reasoning Principles:
- Contrapositive Equivalence: "Antecedent$\rightarrow$Consequent" is equivalent to "$\neg$Consequent$\rightarrow\neg$Antecedent". Neither "denying the antecedent" nor "affirming the consequent" can yield a definite conclusion.
- Transitive Law: "1$\rightarrow$2, 2$\rightarrow$3" is equivalent to "1$\rightarrow$3". The transitive law cannot be applied if the same element appears only on the antecedent or consequent side of all arrows.
- AND and OR Relationships:
- AND Relationship: Indicates a conjunction (and, both...and..., etc.).
2. Naive Logic
- Question Type Characteristics: The question stem provides conditions that require reasoning to derive a conclusion.
- Problem-Solving Methods:
- Use the substitution method when option information is sufficient.
- In more difficult questions, the answer is often among the earlier options.
#### (2) Probabilistic Reasoning
1. Weakening Arguments
- Weakening the Thesis: Directly challenge the thesis by proposing an opposite view or counterexample.
- Weakening the Evidence: Point out flaws or inadequacies in the evidence.
- Breaking the Link: Disrupt the logical connection between the thesis and the evidence.
- Denying the Premise: Negate the necessary conditions for the thesis to hold.
- Causal Inversion: In causal weakening questions, causal inversion is generally the answer.
- Alternative Cause: Weakening in control experiments typically involves identifying an alternative cause.
2. Strengthening Arguments
- Strengthening the Thesis: Explicitly affirm the thesis or provide consistent information.
- Strengthening the Evidence: Provide more robust support for the thesis.
- Establishing a Link: Build a logical "bridge" between the thesis and the evidence.
- Supplementing Premises: Identify indispensable conditions for the thesis to hold.
- Elimination of Alternative Causes: Strengthening in control experiments typically involves eliminating alternative causes.
### II. Problem-Solving Methods and Steps
#### (1) Macro Observation to Determine the Question Type
1. Translational Reasoning: Prioritize translational reasoning when logical connectives are present.
2. Naive Logic: Use naive logic when the question stem contains numerous complex conditions.

3. Probabilistic Reasoning: Identify whether it is a weakening or strengthening question based on the presence of a thesis and evidence in the question stem.

#### (2) Micro Analysis to Lock in Specific Rules

1. Translational Reasoning: Translate the question stem first, then analyze options using reasoning rules.

2. Naive Logic: Reason step-by-step based on the given conditions; use the substitution method when necessary.

3. Probabilistic Reasoning:

- Weakening Arguments: Prioritize direct negation of the conclusion or causal inversion.

- Strengthening Arguments: Prioritize supplementing evidence or establishing logical links.

### III. Common Pitfalls and Tips

1. Translational Reasoning: Remember that "denying the antecedent" and "affirming the consequent" cannot yield definite conclusions.

2. Probabilistic Reasoning:

- In weakening, direct negation of the conclusion and causal inversion are highly effective.

- In strengthening, supplementing evidence and eliminating alternative causes are strongly supportive.

3. Control Experiments: Weakening typically involves alternative causes; strengthening typically involves eliminating alternative causes.

4. Premise Assumptions: Options addressing the "jump" between premises and conclusions in the argument are generally the answer.

Please keep in mind the knowledge points and ways to slove problems for logical judgment that have been given, when answering questions after this:

### E.10 Human Solutions for Logical Judgment in Chinese

下面总结了逻辑判断的核心知识点以及做题方法
### 一、核心知识点
#### （一）必然性推理
1. 翻译推理
- 判断题型：题干或选项出现"如果...那么..., 只有...才..."等典型逻辑关系词。
- 答题技巧：先翻译，后推理。将题干中逻辑关联词所在句子翻译成用箭头推出的关系。
- 翻译原则：
- 谁是必要条件，谁放在箭头后面。如"A是B的必要条件"翻译为"B→A"。
- "...除非... = 除非...否则... = 除非...否则不（不）..."，如"除非A，否则不B"翻译为"B→A"。
- 推理原则：
- 逆否等价："前→后"等价于"否后→否前"，"否前"和"肯后"均推不出确定的结论。
- 传递规律："1→2, 2→3"等价于"1→3"，相同要素如果都在箭头前或后，不能使用传递规律。
- 且与或：
- "且"关系：并列关系（且、并且、和、及、与、同、都、既...又...）。
2. 朴素逻辑
- 题型特点：题干给出一些条件，需要通过推理得出结论。
- 解题方法：
- 选项信息充分时，使用代入法。
- 题目越难，答案越靠前。
#### （二）可能性推理
1. 削弱论证
- 削弱论点：直接对论点发起挑战，提出与之相反的观点或反例。
- 削弱论据：指出论据存在的漏洞或不足之处。
- 切断联系：破坏论点和论据之间的逻辑关联。
- 否定前提：对论点成立的必要条件进行否定。
- 因果倒置：在因果论证的削弱题中，因果倒置基本为答案。
- 另有他因：对比实验的削弱一般为另有他因。
2. 加强论证
- 加强论点：明确肯定论点或者提供与之一致的信息。
- 加强论据：为论点提供更有力的支撑。
- 建立联系：在论点和论据之间搭建逻辑的"桥梁"。
- 补充前提：找到论点成立必不可少的条件。
- 排除他因：对比实验的加强一般为排除他因。
### 二、做题方法与解题步骤
#### （一）宏观观察，确定题型范围
1. 翻译推理：看到关联词，优先考虑翻译推理。
2. 朴素逻辑：题干条件较多且复杂，考虑朴素逻辑。
3. 可能性推理：题干中有论点和论据，判断是削弱还是加强。
#### （二）微观分析，锁定具体规律
1. 翻译推理：先翻译题干，再根据推理规则分析选项。
2. 朴素逻辑：根据题干条件逐步推理，必要时使用代入法。
3. 可能性推理：
- 削弱论证：优先考虑直接否定结论或因果倒置。
- 加强论证：优先考虑补充论据或建立联系。
### 三、易错点与技巧
1. 翻译推理：注意"否前"和"肯后"均推不出确定的结论。
2. 可能性推理： - 削弱中，直接否定结论、因果倒置的削弱力度很强。 - 加强中，补充论据、排除他因的加强力度较强。
3. 对比实验：削弱一般为另有他因，加强一般为排除他因。
4. 前提假设：涉及前提和结论中的跳跃概念的选项基本为答案。
请参考总结的逻辑判断的核心知识点以及做题方法回答接下来的问题：

### E.11 PROMPT FOR MLLMS TO SELF-GENERATE VISUAL REASONING SOLUTIONS IN ENGLISH

Please help me summarize the core knowledge points and problem-solving methods of visual reasoning.

### E.12 PROMPT FOR MLLMS TO SELF-GENERATE VISUAL REASONING SOLUTIONS IN CHINESE

请你帮我总结图形推理的核心知识点以及做题方法

### E.13 PROMPT FOR MLLMS TO SELF-GENERATE DEFINITION JUDGMENT SOLUTIONS IN ENGLISH

Please help me summarize the core knowledge points and problem-solving methods of definition judgment.

### E.14 PROMPT FOR MLLMS TO SELF-GENERATE DEFINITION JUDGMENT SOLUTIONS IN CHINESE

请你帮我总结定义判断的核心知识点以及做题方法

### E.15 PROMPT FOR MLLMS TO SELF-GENERATE DEFINITION ANALOGICAL REASONING IN ENGLISH

Please help me summarize the core knowledge points and problem-solving methods of analogical reasoning.

### E.16 PROMPT FOR MLLMS TO SELF-GENERATE DEFINITION ANALOGICAL REASONING IN CHINESE

请你帮我总结类比推理的核心知识点以及做题方法

### E.17 PROMPT FOR MLLMS TO SELF-GENERATE DEFINITION LOGICAL JUDGMENT IN ENGLISH

Please help me summarize the core knowledge points and problem-solving methods of logical judgment.

### E.18    PROMPT FOR MLLMS TO SELF-GENERATE DEFINITION LOGICAL JUDGMENT IN CHINESE

请你帮我总结逻辑判断的核心知识点以及做题方法

