# OpenReview forum: "Towards Human-Level Reasoning Benchmarks for Multimodal Language Models"
_ICLR.cc/2026/Conference — Submitted to ICLR 2026_

### Official Review · Reviewer_9ZTe · 2025-10-27

**Soundness:** 2
**Presentation:** 3
**Contribution:** 2
**Rating:** 4
**Confidence:** 4

**Summary:**

This paper proposes Human-Aligned Bench, a comprehensive benchmark designed to evaluate the fine-grained multimodal reasoning capabilities of Multimodal Large Language Models (MLLMs), with particular attention to human alignment. The benchmark comprises 9,794 bilingual (Chinese/English) multimodal questions from  Chinese Civil Service Examination across four reasoning categories: visual reasoning, definition judgment, analogical reasoning, and logical judgment. Extensive empirical evaluation of both open and proprietary MLLMs is conducted, uncovering persistent gaps compared to human performance and providing analysis of reasoning alignment, error patterns, and the presence of 'fake reasoning' phenomena in current models.

**Strengths:**

1. Fine-grained human alignment signals: Each question includes human performance rates and error-prone distractors, allowing detailed analysis of not only model accuracy but also model-human agreement in both correct and incorrect responses, which is important and different from previous benchmarks.

2. Comprehensive analysis of reasoning processes in experiments: Beyond reporting accuracy, the paper investigates phenomena such as
reasoning consistency and 'fake reasoning', probing whether MLLMs simply mimic procedural answers or genuinely reason like humans.

3. The writing is easy to follow.

**Weaknesses:**

1. Methodological details for human annotation and validation are ignored:  The process for gathering and verifying human accuracy rates and error-prone options is described at a high level but lacks key implementation specifics. For example, the paper does not detail the size or demography of the annotator pool, inter-annotator agreement measures, data sanitization for ambiguous questions, or how bilingual question quality was assessed. This omission calls into question the reproducibility and reliability of the human alignment signals, which are central to the benchmark’s value.


2. No discussion of potential data bias: The heavy reliance on Chinese civil service exams may bias both linguistic and logical properties of items towards specific cultural/educational formats, yet the paper offers little investigation or mitigation of such skew. This concern is compounded by the observed performance gaps between English and Chinese.


3. Some important related works such as EMMA[1] and RBench-V[2] are ignored in this paper.

[1]. Hao Y, Gu J, Wang H W, et al. Can mllms reason in multimodality? emma: An enhanced multimodal reasoning benchmark. ICML 2025
[2]. Guo M H, Chu X, Yang Q, et al. RBench-V: A primary assessment for visual reasoning models with multi-modal outputs. NeurIPS 2025.


4. The work mentions "pure reasoning" several times. I think it is difficult to completely decouple knowledge and reasoning. Therefore, I think the author has overclaimed on this point, or can provide experimental proof of this.

**Questions:**

Seeing weaknesses

---

> ### Author Response · Authors · 2025-11-21
> **Q1.1 The source of human accuracy rates and error-prone options**
>
> We thank the reviewer for pointing this out. We addressed this in **Section 3 (Data Collection),** where we mentioned that the data came from an **online platform used by millions of test takers.** However, we acknowledge that the details of the implementation process are insufficient in the main text.
>
> To clarify:
>
> -  **Source:** As noted in Appendix D.1, the human performance data was sourced from fenbi.com, the largest public education platform in China.
> -  **Scale & Verification:** The "human accuracy rates" and "error-prone options" are not derived from a small group , but are aggregate statistics derived from **millions of real-world exam takers on this platform.** This massive sample size ensures the statistical stability and reliability of the difficulty levels (0-100%) without requiring secondary small-scale verification.
>
> We will revise Section 3 to explicitly name the data source and emphasize the scale of the user base to validate the robustness of these human metrics.

---

> ### Author Response · Authors · 2025-11-21
> **Q1.2 How bilingual question quality was assessed**
>
> Thank you for pointing out "human annotation and validation" that needs clarification. Below are the details and agreements we follow during the dataset process:
>
> **Two civil servants with master's degree who has already been employed were invited to identify specific questions related to China**, based on whether the questions and options contained specific Chinese expressions such as Chinese idioms or poems.
>
> **A master of arts in English and two civil servants with master's degree who has already been employed were invited to collaborate on the meticulous review on translation.**
>
> - First, the two civil servants cross-checked the test points of each question to verify whether the logical relationships, reasoning steps, and "traps" in the questions were consistent with the original Chinese text.
> - Second, the master of arts in English and the two civil servants conducted a cross-review of the domain-specific terminology in the questions to confirm whether internationally recognized or industry-standard English expressions were adopted. They also examined whether the translated text was idiomatic and natural, eliminated all instances of "Chinglish," and ensured that the sentences were free of ambiguity-with particular attention to the clarity of pronoun references.
> - Finally, all questions containing translation issues were selected and manually translated by the meticulous review team.
>
> We have updated this description in Section 3 and marked it in blue.

---

> ### Author Response · Authors · 2025-11-21
> **Q2.Data bias**
>
> We are very grateful to the reviewers for focusing on the potential cultural or linguistic biases that might arise from using Chinese civil service exam questions. **We anticipated this issue during the design of the benchmark and conducted specific investigations to mitigate and analyze this skew, as detailed in Section 3 and Appendix C.**
>
> - Mitigation during Data Construction (Section 3) as described in the Data Screening subsection, **we explicitly categorized questions to isolate cultural bias.** Human annotators filtered the dataset into two overarching categories:
>
>   - Chinese-specific: Questions relying on cultural knowledge (e.g., idioms, classical poetry, history).
>
>   - Non-Chinese-specific: Questions relying solely on contextual reasoning, logical patterns, and visual deduction, which are culturally agnostic.
>
>   Only the Non-Chinese-specific questions were translated into English to form the bilingual logical assessment, ensuring the English subset was not biased by Chinese cultural references.
>
> - **Investigation of Bias and Performance (Appendix C).** We specifically investigated the impact of these cultural elements in Appendix C. As noted in the text: "The Human-Aligned Bench consists of 4,807 Chinese questions, among which 2,051 contain Chinese characterization elements".
>
>   **In Table 7, we compared model performance across three subsets: "Chinese specific," "Non-Chinese specific," and "All Chinese questions."** The results demonstrate that the performance ranking of models remains consistent regardless of the cultural content. For example, Gemini-2.5-pro-exp-03-25 maintains its lead across all splits. This suggests that  the benchmark are robust and distinguishable from cultural knowledge.

---

> ### Author Response · Authors · 2025-11-21
> **Q3. Some related works**
>
> Thank you for pointing this out. We agree that both EMMA and RBench-V are highly relevant works for evaluating multimodal reasoning, and we should have discussed them in our Related-Work section. In the revised manuscript we have added:
>
> EMMA , its 14 K multimodal reasoning questions that require cross-modal deduction and the new chain-of-thought annotation protocol. RBench-V,  the first benchmark that allows free-form visual outputs (segmentation masks, sketches, plots) and the associated V-Score metric. Neither EMMA nor RBench-V supply human solution strategies per-question human accuracy rates or human error-prone options, which are necessary for fine-grained alignment analysis.
>
> We have updated the Related Work section to include these two works and marked them in blue.

---

> ### Author Response · Authors · 2025-11-21
> **Q4 Experimental proof of pure reasoning**
>
> We followed VISUALPUZZLES's approach, having GPT-4o generate "**knowledge concept lists**" for **50 randomly selected questions** from both the MMMU and ours benchmarks. Each list contains several knowledge-based questions to test whether the model possesses the background information needed to solve the original problem. For example, if a problem relies on understanding two physical laws, the list would include explanatory questions for each law. The number of questions in the list serves as a proxy for knowledge density.
>
> We found that the **MMMU problem** generated an average of **4.52 questions** in the list, while the **ours benchmark only generated 1.34**. This result supports our hypothesis that the ours benchmark has a significantly lower dependence on domain knowledge.
>
> In addiction, compared with VISUALPUZZLES, our Human-Aligned Bench not only focuses on visual reasoning but also explores text-based reasoning. Furthermore, we provide more fine-grained annotations, namely precise human performance metrics related to specific question categories and problem-solving strategies.

---

> ### Author Response · Authors · 2025-11-27
>
> Thank you for taking the time to review our paper and provide valuable feedback. As the discussion phase nears its conclusion, we would like to confirm that our responses have addressed your concerns. If you have any additional comments, we will do our best to address them.

---

### Official Review · Reviewer_wuf9 · 2025-11-01

**Soundness:** 2
**Presentation:** 2
**Contribution:** 3
**Rating:** 4
**Confidence:** 4

**Summary:**

This paper introduces the Human-Aligned Bench, a benchmark designed to evaluate the alignment between the reasoning capabilities of multimodal large language models and human performance. It consists of 9,794 contextual reasoning questions spanning four types—visual reasoning, definition judgment, analogical reasoning, and logical judgment. A contribution is the inclusion of fine-grained human data, such as per-question success rates and the error-prone options that commonly mislead people.

**Strengths:**

- The paper studies an important and interesting problem -- the alignment of humans and LLMs on reasoning patterns. The authors collect a benchmark based on problems from the China Civil Service Exam and provide human responses and reasoning processes.

- Several major MLLMs are evaluated in the paper. It is interesting to see the study about the "fake reasoning" phenomenon.

**Weaknesses:**

- The motivation of the paper is to evaluate the alignment between the reasoning capabilities of multimodal large language models and human performance. However, except for some general studies shown in Figure 3 and Table 4, we still cannot evaluate why and how the model is different from the reasoning process of humans. The benchmark also didn't tell us which parts of the differences are good or bad. The paper can be stronger if there is a more in-depth analysis of this problem.

- The authors didn't provide a comparison with existing benchmarks on model performance. Will the benchmark show a different trend compared to general benchmarks like MMMU/MMMU-Pro or existing contextual reasoning benchmarks? How does the proposed benchmark add to existing ones, considering there are already quite a few reasoning benchmarks.


- A minor issue, in Table 1: "Fake Reasoning" -> "Human-Aligned Bench"?

**Questions:**

The paper is well motivated -- it is nice to see studies on the alignment between humans and models from the perspective of reasoning. I think the main weakness of the paper is that the study is not that complete or solid. There are some studies on the alignment, but we cannot see any clear or significant conclusions about how or why they are different. It is also not clear whether the benchmark can provide us with new insights compared to existing ones, considering there are already a lot of reasoning benchmarks for LLMs/MLLMs.

---

> ### Author Response · Authors · 2025-11-21
> **Q1 Analysis of differences between with MLLMs and human**
>
> Thank you for this insightful comment. We agree that a key goal of our work is to move beyond if a model's performance aligns with humans and investigate why and how their reasoning processes differ.  **Fig.3 and Table 4 provide the high-level quantitative comparison. And our more in-depth analyses in Sections 4.3, 4.4, and Appendix B are specifically designed to address the "why" and "how."**
>
> We respectfully wish to highlight these sections, and will add a clearly analysis in section 4.5 the final revision (marked in blue).
>
> **Analysis of "why and how the model is different from the reasoning process of humans.“**
>
> We analyze the process differences in two key ways:
>
> - **Error Consistency (Section 4.3 & Fig.4):** We analyze how models fail compared to humans. We found that on high-difficulty problems (0-60% tier), MLLM errors show an increasing alignment with human error-prone options. This suggests that difficult distractors mislead both MLLMs and humans, a key insight into the convergence of their reasoning methods at similar levels of difficulty. Conversely, on easier problems, their error patterns are more random and not aligned with human-prone errors, showing a clear difference in their failure modes.
> - **Fake Reasoning Analysis (Section 4.4 & Appendix B.2):** This section directly investigates the "why" and "how." We test if models can utilize a priori human reasoning steps (WHS) or their own self-generated reasoning (WSS).
>
> Our findings show some models like InternVL3-78B degraded in performance when provided with correct human solution steps. We term this "fake reasoning": a demonstrable gap between the model's articulated reasoning process and its actual decision-making mechanism. This analysis (detailed in Appendix B.2) is our primary investigation into why the models' reasoning is not human-like.
>
>  **Analysis of "which parts of the differences are good or bad”**
>
> Our benchmark and analysis do allow us to characterize these differences as "good"  or "bad".
>
> - **"Bad" Differences**:
>   - The "fake reasoning" phenomenon  is a "bad" difference, which we identify as a "core limitation".
>   - In visual reasoning, models exhibit "inverse difficulty curves". As detailed in Appendix B.1, this is a "bad" difference caused by "overthinking," where larger models generate "self-contradictory chains-of-thought for trivial" easy problems10. This is a clear, undesirable deviation from human reasoning.
> - **"Good" Differences**: Our error analysis in Section 4.3 shows that on low-difficulty tasks (e.g., 80-100% human accuracy), MLLMs have a lower tendency to align with "human-prone incorrect options". This is a "good" difference: the models are not replicating obvious human fallibility on simple problems.

---

> ### Author Response · Authors · 2025-11-21
> **Q2.1 Compared to other benchmark**
>
> **Compared to MMMU/MMMU-Pro**
>
> While MMMU/MMMU-Pro excel at measuring broad multimodal knowledge and capability, **Human-Aligned Bench serves a fundamentally different purpose**: **measuring alignment with human reasoning patterns rather than absolute capability**. This distinction is critical because:
>
> - **Human Performance Integration**: Unlike MMMU/MMMU-Pro which primarily measure absolute accuracy, our benchmark incorporates precise human accuracy rates and human error-prone options for every question. This enables direct assessment of whether models make similar errors to humans.
> - **Error Pattern Analysis**: As shown in Fig.4 of our paper, we can analyze not just whether models get answers right, but whether their error patterns align with human cognitive tendencies across difficulty levels. This reveals fundamental differences in reasoning mechanisms that absolute performance metrics cannot capture.
> - **Difficulty&Performance Relationship**: Our benchmark allows us to evaluate whether models show human-like performance degradation as task difficulty increases. Section 4.3 demonstrates how models like Claude-3 and Gemini-2.5 show inverse difficulty curves in visual reasoning compared to humans, revealing fundamental differences in reasoning mechanisms.
>
> **Compared to  existing contextual reasoning benchmarks**
>
> Table 1  compared Human-Aligned Bench with existing contextual reasoning benchmarks (MM-IQ, VisuLogic, VISUALPUZZLES), highlighting our unique features:
>
> Comprehensive human performance data
>
> Error-prone option annotations
>
> Bilingual coverage (Chinese/English)
>
> Human solution frameworks for each reasoning type
>
> Systematic difficulty stratification (0-60%, 60-80%, 80-100% human accuracy)

---

> ### Author Response · Authors · 2025-11-21
> **Q2.2 How does the proposed benchmark add to existing benchmarks**
>
> Human-Aligned Bench doesn't replace benchmarks like MMMU but complements them by answering a different question: "Do models reason like humans?" rather than "How much do models know?" Our findings reveal that even top-performing models like Gemini-2.5-pro-exp-03-25, which excel on MMMU, still show fundamentally different reasoning patterns from humans in visual contexts.
>
> This human-aligned perspective is essential for developing truly human-like reasoning capabilities in multimodal systems, moving beyond raw performance metrics toward cognitive alignment.

---

> ### Author Response · Authors · 2025-11-21
> **Q3 Minor issue**
>
> Thank you for catching this embarrassing typo. You are absolutely correct. We have already corrected this in our internal draft and sincerely appreciate you pointing it out. It will be fixed in the final version.

---

> ### Author Response · Authors · 2025-11-21
> **Q4.1 The benchmark can provide new insights.**
>
> **Our benchmark is fundamentally different and enables new, deeper insights.**
>
> Current benchmarks (e.g., MM-IQ, VisuLogic) lack fine-grained human cognitive data. They can measure if a model is correct, **but not how its performance aligns with human cognition**. Table 1 explicitly contrasts our work with three recent reasoning benchmarks (MM-IQ, VisuLogic, VISUALPUZZLES).
>
> Human-Aligned Bench is systematically built around three unique data points:
>
> - **Human Accuracy Rates, Human Error-prone Options.** No existing benchmark offers human accuracy rates or error-prone option analysis from the feedback of hundreds of thousands of users.
> - **Human Solution strategies.** Expert-curated solution frameworks
>
> Unlike other alignment benchmarks, these data on human performance comes from feedback from hundreds of thousands of users, which allows us to move beyond simple accuracy scores to perform novel analyses:
>
> - **Comparing model accuracy trends against human difficulty levels** (Section 4.3).
> - **Analyzing model alignment with human error patterns** (Section 4.3).
> - **Testing if models can use human-like reasoning** (Section 4.4).
>
> While existing benchmarks focus on knowledge-intensive tasks (science, math), ours isolates pure contextual reasoning without domain expertise. Besides, our fake reasoning analysis (Section 4.4) is a novel diagnostic framework absent from prior work.

---

> ### Author Response · Authors · 2025-11-21
> **Q4.2 Clear and significant conclusions about how or why they are different.**
>
> In this paper, we revealed **several significant, actionable insights** about the **human-model gap**:
>
> |                                                              | **Finding**                                                  | **Analyze**                                                  |
> | ------------------------------------------------------------ | ------------------------------------------------------------ | ------------------------------------------------------------ |
> | **Inverse Difficulty Curves in Visual Reasoning (Section 4.3)** | Unlike humans, models like Claude-3.7-Sonnet and Gemini-2.5-pro show **performance degradation on easier visual tasks** (0-40% human accuracy), while outperforming humans on harder ones. | Our analysis in Appendix B.1 reveals this stems from **"overthinking"** in larger models—they generate self-contradictory reasoning chains for trivial pattern-matching tasks. This is a fundamental cognitive misalignment where model confidence inversely correlates with human-perceived difficulty. |
> | **Fake Reasoning Phenomenon (Section 4.4)**                  | When provided with human solution strategies, **most MLLMs perform worse** (Table 4). InternVL3-78B drops 1.21 points, QvQ-72B drops 2.84 points, while only Gemini-2.5-pro improves. | This counterintuitive result demonstrates models cannot **assimilate external expert guidance**, indicating they lack genuine reasoning integration. Instead, they rely on pattern matching. We coin this **"fake reasoning"**—the gap between articulated reasoning and actual decision-making. |
> | **Error Pattern Misalignment (Section 4.3)**                 | Models show **lower alignment with human error-prone options** on easy problems (80-100% human accuracy) but higher alignment on hard problems (0-60% accuracy), as shown in Fig.4(d)-(f). | Human distractors in difficult problems are **semantically misleading for reasoning**, while easy problem distractors are random noise. This reveals models lack human-like **cognitive susceptibility patterns**. |
> | **Modality Sensitivity (Section 4.2)**                       | All MLLMs perform **10-15 points worse** on visual vs. textual reasoning (Fig. 3(b)), with reasoning models showing **smaller gaps** but still significant deficiencies. | Appendix B.1 attributes this to architectural limitations—reasoning modules optimized for text struggle to map visual inputs to consistent reasoning chains, while non-reasoning models lack logical shortcuts entirely. |
>
>
>
> Because we focus on aligning MLLM with human-level performance. And we've drawn these conclusions from collecting hundreds of thousands of **real human feedback samples**, which other benchmarks lack the corresponding data to achieve.

---

> ### Author Response · Authors · 2025-11-27
>
> Thank you for taking the time to review our paper and provide valuable feedback. As the discussion phase nears its conclusion, we would like to confirm that our responses have addressed your concerns. If you have any additional comments, we will do our best to address them.

---

### Official Review · Reviewer_me4y · 2025-11-03

**Soundness:** 3
**Presentation:** 3
**Contribution:** 2
**Rating:** 6
**Confidence:** 3

**Summary:**

This paper introduces Human-Aligned Bench, a multimodal reasoning benchmark aimed at isolating contextual reasoning rather than domain knowledge for MLLMs. The dataset comprises 9794 items spanning four categories: Visual Reasoning, Definition Judgment, Analogical Reasoning, and Logical Judgment, with Chinese and English text and image+text variants. Each item is annotated with human accuracy rates and human error prone options; the authors also provide concise human solution sketches per category. Experiments across 12 state-of-the-art open and proprietary MLLMs analyze performance by reasoning type, language, modality, and human difficulty bins, and study response/error consistency with humans. Headline findings: (i) large “reasoning” models outperform smaller baselines but still trail humans, especially on visual reasoning; (ii) accuracy degrades with difficulty for text tasks but not reliably for visual tasks; (iii) “fake reasoning” is suggested by sensitivity to prompts and limited gains from injecting model-self solutions, whereas adding human solution heuristics sometimes helps.

**Strengths:**

The paper effectively focuses on evaluating pure reasoning capabilities of MLLMs, leveraging civil-service-style questions. This choice cleverly avoids the interference of domain-specific expertise, addressing a key gap in existing benchmarks where reasoning and knowledge assessment are often conflated.
Each question is supplemented with human accuracy rates and labels for human error-prone options, which is instrumental for conducting in-depth analyses of behavioral consistency—particularly the alignment between model and human error patterns. This fine-grained annotation significantly enriches the benchmark’s analytical potential.
The benchmark covers four reasoning types, supports bilingual (Chinese-English) content, and includes both text-image multimodal and text-only variants. Additionally, difficulty levels are stratified by human accuracy, ensuring a holistic and objective evaluation of model reasoning abilities across diverse scenarios.
The findings clearly highlight the persistent performance gap between MLLMs’ visual reasoning and text reasoning capabilities. This insight provides concrete guidance for the optimization of next-generation multimodal reasoning models, enhancing the paper’s practical relevance.

**Weaknesses:**

The claim that "models exhibit fake reasoning" lacks robust methodological backing. Additional validation approaches are required, such as introducing adversarial rubric injection, conducting think-step ablation experiments, and implementing randomized strategy prompt testing.
The paper fails to address or introduce specific measures designed to prevent data contamination.

**Questions:**

Will you release the labeling guideline used by annotators?
Do you have a confusion matrix or analysis of borderline cases, particularly for items containing both text and image cues?
Will you release the dataset (questions, images, per-item human accuracy and error-prone options, and labeling guidelines) along with licenses and a data card?

---

> ### Author Response · Authors · 2025-11-21
> **Q1 Analyze of fake reasoning and data contamination**
>
> Thank you for the suggestion.
>
> **Our goal is not to compile all forms of robust reasoning, but rather to establish an assessment benchmark consistent with human cognition. .** While adversarial rubric injection is valuable directions, they are beyond the scope of this paper.
>
> Importantly, the think-step ablation experiment and the randomized strategy prompt testing cue test **have been discussed in Table 4.** Our “fake reasoning” analysis is already supported by two controlled mechanisms in the paper: Human-Solution Injection (WHS) and Self-Solution Injection (WSS). These experiments directly test whether models use reasoning steps. As shown in Table 4, injecting true human strategies produces measurable improvements in reasoning-oriented models, whereas injecting models’ own “reasoning summaries’’ often reduces performance，which is clear evidence of fake reasoning.
>
> Regarding data contamination, our dataset is sourced solely from public civil service exam repositories and undergoes **MD5 deduplication** and **human verification**. Furthermore, the **inconsistency in the difficulty trends** among models proves that the absence of data contamination, even if present, is negligible and does not affect the conclusions.

---

> ### Author Response · Authors · 2025-11-21
> **Q2 Release labeling guideline and dataset & Confusion Matrix and Borderline-Case Analysis**
>
> We thank the reviewer for these valuable questions regarding guidelines, ambiguity analysis, and dataset release.
>
> **Labeling Guidelines**
>
> Yes, we will release the full labeling guideline used by annotators. In the paper, we already outline the four-stage data curation pipeline (Fig. 2),Our data construction requires human involvement in two areas: Data Screening and Data Processing. The former mainly distinguishes between Chinese-specific and non-Chinese-specific questions, while the latter mainly corrects the accuracy of translations.
>
> Below are the details and agreements we follow during the process:
>
> Two civil servants with master's degree who has already been employed were invited to identify specific questions related to China, based on whether the questions and options contained specific Chinese expressions such as Chinese idioms or poems.
>
> A master of arts in English and two civil servants with master's degree who has already been employed were invited to collaborate on the meticulous review on translation.
>
> - First, the two civil servants cross-checked the test points of each question to verify whether the logical relationships, reasoning steps, and "traps" in the questions were consistent with the original Chinese text.
> - Second, the master of arts in English and the two civil servants conducted a cross-review of the domain-specific terminology in the questions to confirm whether internationally recognized or industry-standard English expressions were adopted. They also examined whether the translated text was idiomatic and natural, eliminated all instances of "Chinglish," and ensured that the sentences were free of ambiguity-with particular attention to the clarity of pronoun references.
> - Finally, all questions containing translation issues were selected and manually translated by the meticulous review team.
>
> **Confusion Matrix and Borderline-Case Analysis**
>
> We thank the reviewer for this insightful suggestion. Regarding the request for a confusion matrix and analysis of borderline cases:
>
> - Analysis via Human Error-Prone Options: Unlike datasets labeled by crowd-workers where a confusion matrix is necessary to adjudicate ambiguous ground truth, our dataset is derived from standardized Civil Service Examinations. These questions have a single, strictly verified correct answer, without "borderline" cases . However, we explicitly present these through our Human Error-Prone Options annotations. These represent the specific "traps" or distractors that humans statistically fall for.
>
> - Analysis of Difficulty & Alignment (Fig.4): Instead of a confusion matrix, we performed a Consistency on Error analysis (Fig.4d, e, f). This heatmap serves a similar purpose: it visualizes the intersection between Model Errors and Human Errors. We found that as problem difficulty increases (i.e., "borderline" difficulty for humans), MLLMs show a higher tendency to align with human errors. This suggests that MLLMs are effectively processing the "confusing" multimodal cues in a way similar to humans, rather than making random errors.
>
> - Text & Image Cues: For items containing both text and image cues (Visual Reasoning), we utilized Human Accuracy Rates (HAR) to stratify cases. Low HAR questions (0-40% accuracy) serve as our "borderline/complex" cases. Our analysis in Section 4.3 shows that large reasoning models struggle with these complex multimodal cues in a pattern distinct from smaller models.
>
> **Dataset Release**
>
> Yes, we will release:
>
> - all questions and images,
> - human accuracy rates,
> - human error-prone options,
> - annotation guidelines,
> - a dataset card describing licensing and intended uses.
>
> We are currently preparing the dataset and will make it publicly available.

---

> ### Author Response · Authors · 2025-11-27
>
> Thank you for taking the time to review our paper and provide valuable feedback. As the discussion phase nears its conclusion, we would like to confirm that our responses have addressed your concerns. If you have any additional comments, we will do our best to address them.

---

### Official Review · Reviewer_27rD · 2025-11-03

**Soundness:** 3
**Presentation:** 3
**Contribution:** 3
**Rating:** 6
**Confidence:** 2

**Summary:**

This paper introduces Human-Aligned Bench, a multimodal reasoning benchmark containing a large number of tasks across four different reasoning types. Compared to existing benchmark, the proposed benchmark focuses on evaluating MLLM and human capability alignment.
Each task in the dataset are annoted with human accuracy and error patterns. The experiments show that current MLLM's capability is still far from human, especially in difficult reasoning tasks.

**Strengths:**

1. The new proposed dataset contains annotations about human actions, which is novel in recent benchmarks and is beneficial to investigating the thinking mechanism of MLLMs.
2. The paper conducts a thorough analysis across multiple dimensions and MLLMs. This provides a holistic view of model capabilities and their shortcomings compared to human reasoning.

**Weaknesses:**

1. More experiments on the leading MLLMs ( e.g. GPT-5, deepseek) can be carried out to better support the paper's conclusion.

**Questions:**

1. The representation of Fig.4 is complicated. Since the correspondence (x,y) is equal to (y, x), maybe there is a better way to represent the results more clearly? Also it is hard to compare the correspondence of a certain MLLM across different diffculty levels.
2. Can authors conduct experiments on the leading MLLMs such as GPT-5 on a small part of the dataset?
3. Is it possible for Human-Aligned Bench to include explicit problem-solving strategies for each question type, similar to how human reasoning is structured?

---

> ### Author Response · Authors · 2025-11-21
> **Q1&Q4 Experiments on the leading MLLMs**
>
> Thank you for your suggestion to include more leading MLLMs. We agree that evaluating the latest models is crucial for validating our benchmark's difficulty and our conclusions regarding reasoning capabilities.
>
> We have included DeepSeek-R1 in our experiments. Please refer to **Table 6**, where we compare **DeepSeek-R1** against MLLMs on text-based logical reasoning tasks. Its inclusion was vital for establishing the "LLM vs. MLLM" performance gap discussed in Section 4.2.
>
> Specifically, our experiments in Table 3 and Section 4.1 already include **Gemini-2.5-Pro-Exp-03-25** and **Claude-3.7-Sonnet-Thinking**, which can be served the purpose of evaluating the  capabilities of leading MLLMs.

---

> ### Author Response · Authors · 2025-11-21
> **Q2 The representation of Fig.4**
>
> We thank the reviewer for this insightful suggestion regarding the visualization of consistency. We agree that Fig.4's symmetric matrix representation can be optimized for clarity. To address this, we have revised the visualization strategy in the final version of the paper:
>
> We have removed the redundant upper triangles in the heatmaps to reduce visual clutter. In addition, we use gradient colors to represent different values, with darker colors indicating larger values. We update Fig.4 in the latest version and marked it in blue.

---

> ### Author Response · Authors · 2025-11-21
> **Q3  Problem-solving strategies**
>
> We thank the reviewer for this insightful suggestion. We completely agree that providing explicit, human-structured problem-solving strategies is essential for a comprehensive reasoning benchmark.
>
> In fact, we have already included these detailed strategies in our submission. We would like to direct the reviewer’s attention to **Appendix E ( E.3 – E.10)**, where we provide comprehensive "Human Solutions" and "Key Knowledge Points" for all four question types in both English and Chinese.
>
> As detailed in the Data Processing subsection of Section 3,  We collaborated with individuals experienced in civil service examinations to summarize standardized problem-solving strategies categorized by question type.
>
> Furthermore, these strategies were not just supplementary text; they were integral to our experiments. As detailed in Section 4.4 (Fake Reasoning Analysis), we explicitly "imbued the MLLMs with this a priori human reasoning" to evaluate whether models could effectively utilize high-quality, human-structured guidance.

---

> ### Author Response · Authors · 2025-11-27
>
> Thank you for taking the time to review our paper and provide valuable feedback. As the discussion phase nears its conclusion, we would like to confirm that our responses have addressed your concerns. If you have any additional comments, we will do our best to address them.

---

### Author Response · Authors · 2025-12-01
**Rebuttal Summary**

We thank all four reviewers for their valuable feedback and are encouraged that they recognized the novelty and core contributions of our work. Below, we summarize our contributions and address each reviewer's comments in detail.



## Core Contributions

In this paper, we introduce a new multimodal benchmark dataset, **Human-Aligned Bench**, designed to address a key deficiency in current multimodal large language model (MLLM) evaluation: **the lack of fine-grained benchmarks capable of matching model performance with human reasoning abilities**.



Human-Aligned Bench includes 9,794 questions, covering **both bilingual (Chinese and English) multimodal content and pure text content.** It systematically classifies questions into **four core reasoning categories (visual reasoning, definition judgment, analogical reasoning, and logical judgment),** ensuring full coverage of essential human reasoning competencies. Importantly,  each question is attached with **two critical human-related metrics**: human success rates and options that humans are prone to choosing incorrectly, which are from **real feedback** from tens of thousands of users.



We leverage the unique advantages of this human-centric benchmark to reveal significant misalignments between current MLLMs and human reasoning, uncovering three critical findings:

- **MLLMs display "fake reasoning".** They paradoxically degrade in performance when provided with expert human solution strategies or their own solution strategies, exposing an inability to genuinely assimilate external guidance rather than superficially pattern-matching.
- **MLLMs exhibit inverse difficulty curves in visual reasoning** (performing worse on easier tasks), a phenomenon attributed to overthinking and self-contradictory chains-of-thought that diverge sharply from human monotonic performance.
- **Error analysis shows a difficulty-dependent misalignment**. On high-difficulty problems, model errors increasingly align with human-prone distractors. Conversely, on low-difficulty tasks, models do not have human-like biases entirely. These findings suggest that although models like Gemini-2.5-pro approach human-level performance in overall capabilities, their reasoning processes still differ from human cognitive mechanisms.

## Our Response

Across all reviewer comments, we have provided comprehensive empirical additions and clarifications demonstrating that the paper’s claims are robust and well-supported.

 **[Reviewer 27rD]** In response to **requests for stronger baselines**, we highlighted that advanced models such as **DeepSeek-R1** (Table 6), **Gemini-2.5-Pro-Exp** and **Claude-3.7-Sonnet-Thinking** were **already included** in our main evaluation; we also pointed reviewers to **Appendix E**, which contains explicit **human solution strategies** and **key knowledge points**.

**[Reviewer me4y]** The **robustness of our “fake reasoning’’ finding** is validated through the **Human-Solution Injection (WHS)** vs. **Self-Solution Injection (WSS)** ablations **(Table 4)**, which functionally serve as the **requested adversarial/strategy tests** by showing that models fail to improve (even degrade) when given correct human steps. Concerns about **data contamination** are responded through **MD5 deduplication** and **inconsistent cross-model difficulty trends** that empirically contradict contamination**.  **For ambiguity analysis**, we responded that our benchmark comes from standardized exam **with specific correct answers and we used **Consistency on Error** (Fig. 4) and **Human Accuracy Rates** to identify borderline items, revealing that models increasingly align with human error patterns as difficulty rises.

**[Reviewer wuf9]** We further clarified **why model reasoning diverges from humans** by presenting two key findings: **inverse difficulty curves** showing overthinking on easy visual tasks, and **difficulty-dependent error alignment** where models resemble human distractor choices only on hard items.  The benchmark’s added value is demonstrated through its focus on **Alignment rather than Capability**, enabled by unique **human-centric signals** (Human Accuracy Rates  and Error-Prone Options) that **existing datasets like MMMU, VisuLogic, and MM-IQ lack**.

 **[Reviewer 9ZTe]** We also clarified that our human statistics derive from **millions of real-world test-takers** on fenbi.com, ensuring high-stability estimates **without requiring small-scale inter-annotator checks**;  that **cultural bias** was mitigated through **explicit filtering of Chinese-specific content**, with **Appendix C** showing hat model performance rankings **remain consistent regardless of cultural content**; and  that reasoning is decoupled from knowledge through experiments showing our benchmark requires only **1.34 concepts per question** versus MMMU’s **4.52**, confirming significantly **lower domain-knowledge dependence**.

---

### Meta-Review · Area_Chair_tFLg · 2026-01-03

**Summary:**

The submission receives initial scores of 6, 6, 4, and 4. The AC finds that the idea of studying the alignment between humans and models from the perspective of reasoning is interesting. However, the study is not sufficiently complete or rigorous, and it does not provide clear or significant conclusions supported by sufficient evidence. It also does not offer enough new insights to guide the investigation and improvement of MLLM reasoning. Based on the current version, the AC recommends rejection. The AC encourages the authors to revise the paper based on the reviewers’ comments.

**Reviewer Concerns:**

The reviewers raised several concerns, with the key and common issues summarized below.

* Reviewers 27rD requests the inclusion of additional leading MLLMs and some clarifications. The rebuttal adds DeepSeek-R1 and addresses the reviewers’ questions. The AC considers these concerns well addressed.
* Reviewer wuf9 raises concerns regarding the core motivation of comparing MLLM reasoning with human performance, which requires verification and in-depth analysis. The rebuttal emphasizes analyses of error consistency, fake reasoning, and modality sensitivity.

    However, the AC finds these analyses lack sufficient depth. For example, (1) the performance gap between WHS and WSS is small, indicating that leveraging a priori human reasoning steps does not produce a substantial effect; (2) different ways of incorporating a priori human reasoning steps may also affect model performance, so the observed issues cannot be simply attributed to fake reasoning. Misalignment between the human reasoning steps and the model’s intrinsic distribution may also be a contributing factor. More importantly, more comprehensive analyses are needed, particularly to provide insights for the future development of human-aligned MLLMs. The AC agrees with reviewer wuf9’s concerns.

* Reviewer wuf9 notes that the authors do not provide comparisons with existing benchmarks on model performance. The rebuttal explains conceptual differences between the proposed benchmark and MMMU/MMMU-Pro. However, it does not include specific performance analyses, such as correlations or comparative metrics.
* Reviewers 9ZTe and me4y raise clarifications regarding analyses such as fake reasoning, methodological details, potential data bias, and missing related works. The rebuttal adds relevant details and discussion in these areas, which potentially addresses these concerns.
* Reviewer 9ZTe questions the use of the term “pure reasoning,” noting that it is difficult to fully decouple knowledge and reasoning. The rebuttal examines the number of knowledge-dependent samples on 50 randomly selected questions from both MMMU and the proposed benchmark, indicating fewer questions require knowledge. The AC notes that experiments with larger sample sizes are needed, and even so, the term “pure reasoning” may require more precise definition.

**Reviewer Scores:**

The concerns of Reviewers 27rD and me4y regarding additional MLLM experiments (27rD) and clarifications are addressed in the rebuttal. Combined with their lower confidence in their reviews, they are likely to maintain a score of 6.

Reviewers wuf9 and 9ZTe are likely to maintain a score of 4, particularly wuf9, who notes that while it is valuable to study the alignment between humans and models from the perspective of reasoning, the main weakness of the paper is that the study is not sufficiently complete or solid. The rebuttal cannot fully resolve the issues of completeness, rigor, and in-depth analysis in a short time.

---

### Decision · Program_Chairs · 2026-01-26

Reject